

# Integrated dataset of deformation measurements in fractured volcanic tuff and meteorological data (Coroglio coastal cliff, Naples, Italy).

Matano Fabio[1], Caccavale Mauro[1-2], Esposito Giuseppe[1-3], Fortelli Alberto[4], Scepi Germana[5], Spano Maria[5], Sacchi Marco[1]

[1]Istituto di Scienze Marine (ISMAR), Consiglio Nazionale delle Ricerche (CNR), Naples, 80133, Italy
[2]Istituto Nazionale di Geofisica e Vulcanologia (INGV), Osservatorio Vesuviano, Naples, 80124, Italy
[3]Istituto di Ricerca per la Protezione Idrogeologica (IRPI), Consiglio Nazionale delle Ricerche (CNR), Rende, 87030, Italy
[4]Centro Interdipartimentale di Ricerca, Laboratorio di Urbanistica e di Pianificazione Territoriale "Raffaele d'Ambrosio (LUPT), Federico II University, Naples, 80132, Italy
[5]Dipartimento di Economia e Statistica, Federico II University, Naples, 80126, Italy

*Correspondence to*: Fabio Matano (fabio.matano@cnr.it)

**Abstract.** Along the coastline of the Campi Flegrei volcanic district, near Naples (Italy), severe retreat processes affect a large part of the coastal cliffs, mainly made of fractured volcanic tuff and pyroclastic deposits. Progressive fracturing and deformation of rocks can lead to hazardous sudden slope failures on coastal cliffs. Among the triggering mechanisms, the most relevant are related to meteorological factors, such as precipitation, freezing conditions, and thermal expansion due to solar heating of rock surfaces. In this paper, we present a database of measurement time series taken over a period of ~ 4 years (Dec 2014-Oct 2018), referred to the deformations of selected tuff blocks in the Coroglio coastal cliff. The monitoring system is implemented on five unstable tuff blocks and is formed by nine crackmeters and two tiltmeters equipped with internal thermometers. The system is coupled with a total weather station, measuring rain, temperature, wind and atmospheric pressure operating since January 2014 up to Dec 2018. A measurement frequency of 10 and 30 minutes has been respectively set for meteorological and deformation sensors. The aim of the measurements is to assess the magnitude and temporal pattern of rock block deformations (fracture opening and block movements) before block failure and their correlation with selected meteorological parameters. The results of a multivariate statistical analysis of the measured time series suggest a close correlation between temperature and deformation trends. The recognized cyclic, sinusoidal changes in the width (opening/closing) of fractures and tuff blocks rotations are ostensibly linked to multiscale (i.e. daily, seasonal and annual) temperature variations. Some trends of cumulative multi-temporal changes have been also recognized. The full databases are freely available online at https://doi.pangaea.de/10.1594/PANGAEA.896000 (Matano et al., 2018) and https://doi.org/10.1594/PANGAEA.899562 (Fortelli et al., 2019).



# 1 Introduction

Rocky coastal cliffs are located at the transition zone between subaerial and marine geomorphic systems. They represent a very dynamic environment influenced by complex geological evolution on both regional and local scales (Emery and Kuhn, 1982; Bird, 2016; Sunamura, 1992). In fact, the evolution of rocky coasts often occurs as a progressive retreat of the cliff landward induced by a complex combination of marine (i.e. wave action) and subaerial processes (i.e. weathering, erosion and mass movement) (Sunamura, 2015). Future cliffs recession could be more intense in the next decades under the ongoing accelerating sea-level rise thought likely to result from global warming (Bray and Hooke, 1997). In this context, sea cliff failures represent a serious hazard for population living in coastal settlements. Severe retreat processes, mainly occurring with landslides and erosion, are affecting, for instance, many of the coastal cliffs forming the Italian rocky coastline in various geological contexts (Iadanza et al., 2009; Furlani et al., 2014). Here, failures may affect cliff formed by carbonate rocks (Andriani and Walsh, 2007; Ferlisi et al., 2012), arenaceous-pelitic or calcareous-pelitic flysch (Budetta et al., 2000; De Vita et al., 2012), sandstones (Sciarra et al., 2015), shales (Raso et al., 2017), or volcanic rocks (Barbano et al., 2014).

The use of systems aimed at monitoring the slope stability is becoming a standard practice to assess and prevent geological and geotechnical hazards, and plan effective actions for hazard analysis and risk mitigation. Several on site cliff monitoring systems are operative in mountainous environments (Pecoraro et al., 2018), as for example in the Swiss Alps (Spillmann et al., 2007), Bohemia (Zvelebill and Moser, 2001), Spain (Janeras et al., 2017) and Northern Apennines (Salvini et al, 2015). These systems have the purpose of detecting and measuring small-scale rock deformations that can be regarded as precursors of slope failures. A more limited number of monitoring systems is used for the analysis of coastal cliffs and early warning purposes (Clark et al., 1996; Cloutier et al., 2015; Devoto et al., 2013). In our research (Matano et al., 2015, 2016; Esposito et al, 2017, 2018; Caputo et al., 2018), we focus on the in situ measurements of deformations affecting rocky blocks that form part of a volcaniclastic coastal cliff located along the coastal sector of the Campi Flegrei, a densely urbanized volcanic area in southern Italy (Fig. 1a). In this area, marine coastal cliffs are engraved in different volcanoclastic deposits (mainly tuff and weakly welded ash and pumice layers), representing, in many cases, erosional relics of volcanic edifices formed during explosive eruptions occurred at Campi Flegre in the last 15 ka. In detail, we present the database of measurements obtained by an integrated monitoring system implemented at the Coroglio coastal tuff cliff, located in the highly urbanized coastal area of Naples (Italy) (Fig. 1a). The main aims of the research are to identify range and patterns of deformation during a potential pre-failure stage of rock blocks, and to assess eventual relationships between meteorological factors (temperature, rain, wind, humidity, atmospheric pressure, etc.) and deformation of rocks. Besides, the statistical analysis of multi-temporal datasets we



present in this study may provide an experimental basis for the appropriate definition of tuff failure early-warning systems set-up.

## 2 Site description

The analyzed sector of the Coroglio cliff (Figs. 1b and 2) is ~ 140 m high and 250 m wide, with aspect towards the SW. Different geological units and slope angles characterize the cliff (Fig. 3). The upper part displays slope angles varying from 35° to 45° and is formed by stiff to loose Holocene pyroclastic deposits (LP unit) about 30 m in thickness, including a 2-3 m thick layer formed by very loose, reworked volcaniclastic deposits and soils at the top of the succession exposed on the cliff surface.

The median part of the cliff is characterized by almost vertical slope and is formed by two tuffaceous units, separated by an angular unconformity. The Neapolitan Yellow Tuff (NYT) formation is a lithified ignimbrite deposit dated at ~15 ka BP (Deino et al., 2004). The NYT represents the upper unit in the outcrop and is formed by alternating coarse-grained, matrix-supported breccia, thin-laminated lapilli beds and massive ash layers. The NTY displays a relatively homogeneous texture with several sub-planar surfaces likely controlled by structural discontinuities, and can be classified as weak to moderately weak rock (Froldi, 2000). The Trentaremi (TTR) formation forms an older tuff cone dated ~ 22.3 ka BP and represents the lower unit, consisting of slightly welded to welded, whitish to yellow, coarse-grained pumiceous fragments embedded in a sandy ash matrix and lapilli beds. Diffused dm-scale vesicles and vacuoles due to differential erosion, markedly controlled by the bedding of the pyroclastic deposits, characterize the TTR rockface. These forms are typically related to the weathering due to salty seawater and wind erosion (Ietto et al., 2015).

The tuff units cropping out at the Coroglio cliff are characterized by a complex system of mostly steep and planar structural discontinuities and fractures (Matano et al., 2016) showing highly variable spacing, well-developed NE–SW and NW–SE directions, and subordinate N-S and E-W trends. The base of the cliff is covered by slope talus breccia (dt), partly occurring directly along the shoreline. These deposits are produced by the frequent failures occurring along the cliff. The geomorphological instability is due to several causes, such as the complex volcano-tectonic evolution, the severe anthropic modifications (i.e. excavations, tunnelling, and so on) of the coastline since Roman times and weathering and erosion processes occurring at the boundary between coastal and marine environment. The last relevant tuff failures occurred around 1990 (Froldi, 2000) along the southern sector of the cliff (Fig. 2). Due to the severe instability conditions, the upper part of the northern sector of the tuff cliff has been subject to reinforcement works (Fig. 2) at the beginning of the 2000s. These consisted in a wire mesh and steel cable network applied to the tuff wall steel reinforced with bars anchored and bolted to the rock.

During the first phase of the research, we have integrated the results of long-range terrestrial laser scanner (TLS) surveys with structural field mapping to obtain a detailed geostructural analysis and classification of the slope (Fig. 4) (Matano et al., 2016). Repeated TLS surveys have been used for obtaining detailed multi-temporal DTM of the cliff (Caputo et al., 2018) that allowed mapping the landslides occurred during 2013-2015 (Fig. 3). In detail, four types of

landslides (rock fall, debris fall, earth flow, and soil slip) have been recognized in the different sectors of the cliff (Fig. 3). These have caused a total eroded/fallen volume of about 3500 $m^3$ of volcanic material (rock and soil) during the 2013-2015 time period. The related short-term retreat rate of the entire cliff has been assessed in about 0.07 m/yr (Caputo et al., 2018). Based on a geomorphological analysis, including cliff inspections carried out by climber geologists and interpretation of the multi-temporal DTMs, an area with several unstable tuff blocks (Fig. 2) have been

recognized in the NYT unit sector of the cliff (Fig. 3). This area has been selected for implementing the monitoring system.

## 3 The monitoring system

Based on the results of the geo-structural and geomorphological surveys, a series of prismatic tuff blocks, bounded by open fractures, have been selected for the instrumental monitoring (Figs. 4 and 5). The accurate detection of structural

discontinuities and morphometric parameters of the selected tuff blocks resulted in the understanding of the possible failure mechanism and their kinematic analysis. Volumes of the selected tuff blocks range between 4 and 15 $m^3$, and the possible failure kinematics are toppling, planar and wedge sliding (Tab. 1).

The installed monitoring system consists of both standard geotechnical instruments, such as crackmeters and tiltmeters equipped with near-rock-surface thermometer, installed in correspondence of the 5 selected unstable tuff blocks, and a

weather station equipped with thermometer, barometer, hygrometer, anemometer and rain gage sensors. The used instrumentation forms hence two linked separate sub-systems: the Coroglio cliff Monitoring System (CC-MoSys) and the Denza Meteorological Station (DeMS) (Fig. 1b).

Specifically, the CC-MoSys is formed by 9 mono-axial electric crackmeters (model BOVIAR BTS LWG 100), 1 external thermistor (model BOVIAR NTC/A 10K) and 2 biaxial tiltmeters (model BOVIAR BIAX-B/T) equipped with

internal thermistors (model BOVIAR NTC/A 10K) (Tab. 2). The geotechnical sensors have been installed across or on the face of the fractures bounding the five unstable tuff blocks in order to provide an accurate monitoring of displacements and rotations. Each sensor is individually cabled to the acquisition unit (model BOVIAR eDAS 16ch; see at https://www.boviar.com/public/pdf/636489456594961401eDAS%2016-32%20CH.pdf), located at the top of the cliff. The DeMSys is formed by a meteorological station (DAVIS, model Vantage Pro2 – wireless; see at

https://www.davisinstruments.com/support/vantage-pro2-wireless-stations/), located less than 1 km away from the cliff





(Fig. 1b), characterized by a wireless Integrated Sensor Suite (ISS) including barometer, hygrometer, pluviometer and thermometer integrated sensors and by an anemometer (Tab. 3). The CC-MoSys tuff monitoring unit records 16 parameters (Tab. 4) with a frequency set up to 30 minutes for 48 daily measures. The weather station unit records 9 parameters (Tab. 5) with a frequency of measurements set up to 10 minutes for 144 daily measures. Continuous data

recording of geotechnical sensors has been active since December 2014 to October 2018, whereas the weather station is still active since January 2014. The meteorological time series analyzed in this work are updated to December 2018. Monitoring system has been regularly controlled and maintained so to ensure a fully accuracy in measures, even if some missing data intervals are present due to the occurrence of some functional interruptions of the acquisition units.

Overall, the monitored parameters include both deformation and meteorological ones:

• Variation of the cracks opening and Displacement (sliding) along tuff cracks,

• Slope angle variations (rotation) of tuff block surfaces,

• Air temperature and Near-rock-surface air temperature,

• Humidity,

• Wind (velocity and direction),

• Barometric pressure,

• Rainfall (rain quantity and rate).

The data acquisition and broadcasting system is fully remotely controlled. Rechargeable batteries and a solar panel provide the power supply for the monitoring system. Twice a day, the measured data are transferred via GSM (Global System for Mobile Communications) network to a master station located in the ISMAR (Istituto di Scienze Marine)

Research Institute, where data are directly stored in a dedicated NAS (Network Attached Storage) system and converted to open access files for our data repository.

## 4 Tuff deformation data

The deformation data consist of measurements of displacement across the fractures bounding tuff blocks along the direction of the crackmeters axis and of both horizontal and vertical angles associated with rotation of the rock blocks

(Tab. 2). The instruments have collected measurements every 30 minutes for a period of about 4 years. These measurements are expressed for each sensor as relative measurements with respect to the initial measurement, considered as a reference zero value. We also measured near-rock-surface air temperature (as a proxy for surface rock temperature) using some thermistors installed at three locations on the cliff (i.e. near the acquisition unit and near the tiltmeter boxes). In this way the CC-MoSys database encompasses a total of 67355 measurements acquired during 1404

30  days (from 3 Dec. 2014 to 6 Oct. 2018). Some missing data intervals are present due to the occurrence of some



functional interruptions of the eDAS acquisition unit; for two sensors (F04F1 and F16F2) missing data intervals are larger (Table 6) due to the occurrence of mass movements on the cliff that damaged the two sensors.

We compute some descriptive statistics for summarizing the different features of the deformation measurement dataset (Tables 6). These statistics are the results of a univariate analysis, describing the distribution of each single variable,

including its central tendency (mean and median) and data dispersion (including minimum and maximum values, quartiles and standard deviation).

Crackmeters data usually show max-min values between 0.48 and 1.98, with the two relevant exceptions of 04F1 and 05F2, respectively with ranges of 5.56 and 4.31. Tiltmeter data show a lower variability as max-min value intervals range between 0.38° and 0.78° (Tab. 6). More in detail, the data distribution analysis show some different trends for the

data of the different sensors. Data of 03F1 sensor shows an asymmetric distribution with modal value around - 1 mm and a higher coda values up to 3 mm. The 04F data show a uniform distribution between -0.1 and 3 mm (Figure 6). The block 05 have a very large range of variability, with a well-defined main value only for F2 sensor (Figure 7a). F1 and F3 sensors show a very high correlation (Figure 7b), suggesting a uniform behavior of the crack in the section covered by them. All the three sensors recorded large distribution of both positive and negative values, indicating a strong

dynamic behavior. On the other hand, the 16F1 sensor shows a mono-modal distribution well centered around 0.2 mm (with an oscillation of ±0.1 mm). The stability of the values across the time could highlight a stable dynamic position reached from the block. The 16F2 sensor worked for a very short time compared with the others. However, the data distribution shows two main values 0.0 and 0.2 mm (Figure 8). The F1 and F2sensors, located on the tuff block 19, show a quite identical behavior. (Figure 9). The similarity in recorded values suggest a uniform behavior of the lateral

crack.

The tiltmeter data has been reported in Fig. 10 and Fig. 11 for 04I2 and 19I3 sensors, respectively. From both plots, the X (horizonal) components recorded prevalently negative variation, while the Y (vertical) components positive one over the entire time interval of the datasets. Those evidences show how the rock block moves and turns during the time and some preferred position (the higher histogram bar). For the 19I3 sensor, a small range of variability has been observed,

about 0.4 degree, and a tiny distribution around 0.3-0.4 degree for the Y component. A similar behavior is shows by component X where the equilibrium position is characterized by angle value between -0.20 and -0.30.

The temperature values distribution of the three thermistors appear coherent and well centered in a range between +13 and +25 Celsius degree (Fig. 12). If we compared those temperature data with air temperature measurements provided by the weather station installed in the vicinity (Fig. 13), time series show as usual for this climatic region a clear

seasonal variation patterns and annual cyclicity with some little differences. In detail, near-rock-surface air temperatures show minimum values from 2.8 to 3.7 °C higher and maximum values from 4.4 to 9.7 °C higher than air temperatures (Tab. 6). Amplified near-rock-surface air temperatures could result in consequent increases in crack deformation forces.

If we consider the time distribution of measured data, both crackmeters and tiltmeters show several evidences of tuff deformation through time. The time series of the measured crack aperture and plane rotation data reveal a deformation pattern characterized by both diurnal and seasonal cyclical deformation (Figs. 14 and 15). Data plot referred to a single year of measurements show a similar repetitive pattern with lower values in late winter and higher values in late

summer (Fig. 16) showing significant correlation with seasonal temperature trends. Some trends of cumulative multi-annual changes, based on about four consecutive years of data (December 2014 to October 2018), can be also recognized. Crackmeters show from +0.1 to +1.2 mm cumulative deformation (Fig. 14), while tiltmeters show from 0.1° to 0.4° cumulative angle variation (Fig. 15).

## 5 Meteorological data

The database of DeMSys measurements (Fortelli et al., 2019) encompasses 262944 measures acquired with a 10 minutes interval during 1826 days spanning from January 2014 to December 2018.

Some missing data intervals are present due to the occurrence of some functional interruptions of the DAVIS acquisition unit; only for the anemometric sensor (Wi_speed and HiWi_spe) missing data intervals are more large (Table 7) due to a technical issue of the sensor during 2018. The descriptive statistics resulting from a univariate

analysis of the acquired dataset, with the exception of wind and gust direction, is shown in Table 7.

We added to the database two numeric parameters that take into account the direction of wind and gust, i.e. wind pressure (Wi-P_norm) and gust pressure (HiWi-P_norm), expressed in $Nm^{-2}$. They are referred to the normal component to the cliff of the pressure exerted by the wind/gust on the rock face. We calculated the normal component to the cliff of the wind velocity by considering the angle between cliff average aspect orientation, that is towards WSW

(i.e. 247.5°) and the wind/gust direction. Winds blowing from direction of 337.5° to 360° and from 0° to 157.5° do not produce pressure on the rock surface as they are sheltered from the cliff itself. Winds blowing from directions between 157.5° and 337.5° produce a normal component of pressure that varies under the cosine of the incidence angle with the cliff (ranging from 0° to 90°). Once calculated the vertical component of the wind/gust, we may calculate the wind/gust pressure normal to the cliff with the simplified formula (ASCE, 2013):

$Pn = 0.613 \, vel^2$                                                                                           (1)

where:

Pn = wind/gust pressure normal to the cliff expressed in $N*m^{-2}$

vel = wind velocity normal to the cliff expressed in $m*s^{-1}$

0.613 = coefficient based on average values of air density and gravitational acceleration, expressed in $kg^{-1}m^{-1}$



### 5.1 Thermal trends

Temperature time series data (Fig. 13) show a clear seasonal cyclicity and inter-annual variation patterns. Air temperature values range between -2.9°C and 33.8°C throughout the five years of measurements.

Temperature average values of the 5-year period have been plotted (Figs. 17 and 18), referring to:

- Average daily values analysis, referred to daily minimum temperature (Tmin) and daily maximum temperature (Tmax) in order to observe the annual trends and detect the most significant surplus or deficit thermal phases (Fig. 17). Diagrams also report interpolation curves relative to minimum and maximum daily values (4th grade polynomial equation); we also computed the difference between the minimum and maximum temperatures.

- Average monthly values analysis, referred to daily monthly minimum (MTmin) and daily monthly maximum
10   values (MTmax) and extremes for both minimum and maximum (Tmin-Tmax) (Fig. 18).

In detail, we have evaluated the thermal increase between late winter minimum and summer maximum and the autumn thermal decrease between summer maximum and late December minimum. From the analysis of 4th grade polynomial equations of annual thermal trend, relative to daily Tmin and Tmax, the following values have been derived:

|  |  |  |  |
|---|---|---|---|
| Tmin(1) (min) = | 8.0°C (05/02) | Tmax (1) (min) = | 12.3°C (04/02) |
| Tmin (max) = | 21.9°C (30/07) | Tmax (max) = | 27.5°C (02/08) |
| Tmin(2) (min) = | 7.9°C (31/12) | Tmax (2) (min) = | 12.2°C (31/12). |

Thermal yearly excursions are therefore equal to:

spring thermal increase:   $\Delta T+$ (min) = Tmin (max) – Tmin(1) (min) = 13.9°C

$\Delta T+$ (max) = Tmax (max) – Tmax(1) (min) = 15.2°C

20   autumn thermal decrease: $\Delta T-$ (min) = Tmin (max) – Tmin(2) (min) = 14.0°C

$\Delta T-$ (max) = Tmax (max) – Tmax(2) (min) = 15.3°C

The main features highlighted by the analysis are the following:

- an annual trend showing a wave characterized by a weak thermal excursion, both for minimum and maximum daily values;

- a summer season with only a few days with daily maximum temperature above the threshold value of 30°C;

- an extremely mild winter season, with only a few days with minimum temperatures below 5°C;

- a weak daily thermal excursion, with values are generally ~ 5°C and only a few days with a daily thermal excursion > 10°C.

In detail, the annual thermal trend in the analyzed period is characterized by three years (2016-18) with values close to
30   the average values of the investigated period, and two years with significant differences during the summer season. In detail, summer 2014 had values significantly lower than the average values, and summer 2015 had values characterized





by a clear thermal surplus during the months of July and August, when the threshold value of 30°C has been exceeded in 22 days. During the winter there weren't any long periods characterized by significantly low temperatures, with the two exceptions of the period between 5 and 15 January 2017, (with values close to zero and minima of -1.6°C) that also caused weak snow falls), and a very cold and snowy phase at the end of February 2018.

## 5.2 Rainfall trends

Rain and rain rate data time series are plotted in Fig. 19. Rainfall is characterized by a considerable irregularity, with sometimes extremely high accumulation values and rates (October 2015, September 2017), alternating with periods of modest rainfall even in seasonal phases in which rainfall is generally abundant (December 2015 and 2016). Rainfall (measured at intervals of 10 minutes) reached a maximum value of 17.6 mm on 12/09/2014, corresponding to a rain rate of 105.6 mm/h. The maximum rain rate of 292.6 mm/h was recorded on 19/09/2016.

We plotted the average values referred to the whole 5-year period of the monthly total rainfall amounts (Fig. 20). In order to evaluate the occurrence of rainfall *surplus* or *deficit* periods, these values are compared with the monthly average values of rainfall, referred to 1872-2005 time period, measured at the Meteorological Observatory (MOUF) of the University of Napoli "Federico II" (Mazzarella, 2006). It is worth noting that the monthly precipitation average values over the analysed 5-years period (2014-2018) tend to converge towards those values of climatological relevance recorded in Naples at MOUF (Mazzarella, 2006), thus reflecting the strong pluviometric characterization of the site. Data analysis highlights some peculiarities in the rainfall conditions that affect the area. The average annual rainfall amount of 759,6 mm is below the amount of climatological reference of 866,0 mm. In detail, the summer rainfall amount is almost equal to MOUF climatologic reference (69,0 mm vs 75,7 mm), while spring (149.3 mm vs 181.4 mm), autumn (281.9 mm vs 317.6 mm) and winter (259.4 mm vs 290.9 mm) rainfall amounts are slightly below MOUF climatologic reference. The winter precipitation deficit is due to strong negative rain anomaly observed at Denza during December 2015 (0.3 mm) and 2016 (9.5 mm). During 2015, a rainfall regime with a large prevalence of dry months was compensated by three very rainy months, January, February and October, the last one characterized by a rainfall amount of 195.5 mm. Year 2017 has been characterized by a severe rainfall *deficit* with a total annual precipitation amount of 536.8 mm; particularly, summer 2017 rainfall amount was of only 4.1 mm, suggesting extremely dry conditions for this season. Instead, year 2018 has been characterized by a total annual rainfall amount of 902.1 mm, slightly above MOUF climatologic values.





### 5.3 Wind trends

Wind and gust speed data time series are plotted in Fig. 21. Wind velocity and gust velocity (both measured at 10-minutes intervals) reached the maximum values of 40.9 knots (11.8 m/s) and 61.7 knots (31.7 m/s), respectively (Tab. 7).

The wind regime indicates considerable consistency, both in terms of anemoscopic regime (wind provenance direction), and in terms of average intensity. The seasonal regimes are characterized by the repetition of the anemometric patterns, highlighting the occurrence of local structural factors that, even if in interaction with the meteorological conditions on a synoptic (cyclonic) scale, are able to control the wind regime in the considered coastal sector. Tables 8 and 9 summarize the annual and monthly average values of average wind speed (m/s), dominant direction of the wind (sector of 22.5°)

and maximum wind gust (m/s). Wind average speed usually can be referred to level 2 in Beaufort scale, while gust maximum speeds can be referred to levels from 9 to 11 in Beaufort scale.

The rose diagram of Fig. 22 shows the frequency distribution of wind and gust directions, which are mainly from SSE, NE e WSW. During spring the synoptic winds blow with greater frequency from the II and IV quadrants, with the SE direction reaching the maximum percentage weight. The direction associated with the maximum average wind speed

values is the SE - SSE directions. Therefore we may infer that the spring season is characterized by the alternation of south-eastern and north-western winds, that are also associated with the greatest average intensities. The polar diagram shows that summer winds blow with a prevalence from the III and IV quadrants, with the WSW direction reaching the maximum percentage weight. The direction associated with the maximum average wind speeds are the westerly ones. It is therefore possible to state that the summer season is dominated by breeze regime winds, typical of the coastal

Mediterranean areas. However, it is worth underlining that the occurrence of SSE relatively high average intensities is not infrequent. In the autumn season winds blow in a well distributed in all sectors, with the SE-SSE direction reaching the maximum percentage weight; also the directions associated with the maximum average wind speeds are the direction SE and SSE. It is therefore possible to state that the autumn season is normally dominated by Scirocco-like anemometric events, with associated very rough sea. During winter season, the polar diagram highlights an

anemoscopic regime very similar to the autumn one. In fact, it may be observed that the synoptic winds blow in a well distributed way involving I, II and IV quadrants, with the SE and NE directions reaching the maximum percentage weight. The provenance directions associated with the maximum average wind speeds are namely SE and SSE.

By considering the angle between wind/gust direction and Coroglio cliff aspect, we have calculated the values of the normal component of the wind/gust pressure with respect to the rock face exposed on the cliff, i.e. wind normal

pressure (Wi-P_norm) and gust normal pressure (HiWi-P_norm). Wind and gust normal pressure reached the maximum values of 185.6 N/m$^2$ and 1400.6 N/m$^2$, respectively.





### 5.4. Humidity and barometric pressure trends

Humidity and barometric pressure data time series are plotted in Fig. 24. Relative humidity values range between 24.7 % and 98.0 % with mean values around 72.8 % (Tab. 7). The monthly, seasonal and annual trends are characterized by similar, repetitive patterns with usually high values between 70-80%. This as a direct consequence of the closeness to
the sea surface, and of forced lifting of air masses due to the orographic factor, leading to a cooling with associated increase in relative humidity. The barometric pressure values range between 983.2 mm and 1036.7 mm with mean values around 1015.6 mm (Tab. 7). The seasonal and annual trends are characterized by similar, repetitive patterns, showing a high variability of values during the six-months cold period and a lower one during warm period, this due to the synoptic scale meteorological perturbations frequently affecting the area.

### 6 Correlation and regression analysis of data

Tuff deformation measurements were collected at 30 minutes intervals whereas the meteorological data were recorded at 10 minutes intervals. Successively, we have analyzed the correlations among the time series of the different parameters. Therefore, we have aggregate the data into 1826 daily records (from 1 Jan. 2014 to 31 Dec 2018). We have referred to the daily average values for all deformation data and partly for meteorological data; in detail, we referred to
the highest daily value for "Rain_rate", "HiWi_spe" and "HiWi_P_norm" variables, to the cumulative daily value for "Rain" variable and to the daily average values for the others. The daily descriptive statistics are shown in Table 10.

In a multivariate perspective, we examine the correlation among tuff deformation parameters and meteorological variables in order to detect possible relationships between the two data sets. We compute the correlation matrix, where the Pearson correlation coefficient is the generic element. This coefficient varies between -1 (maximum negative
correlation) to +1, (maximum positive correlation). Particularly, correlations greater than |0.5| are regarded as highly significant, whereas coefficient between |0.33| and |0.5| are considered slightly significant.

Table 11 shows a part of this matrix related to the correlations. We observe that the highest positive correlations are between **F04F1** and **Temp** (0.73), and anyway all the correlations between **F19I3-X** and the temperature variables are very high. Other two deformation variables (**F04F1, F19I3-Y**) are very positively correlated with all the temperature
variables. On the other hand, the variable **F16F2** shows a high negative correlation with two meteorological variables related to the measure of wind **Wi-P_norm (-0.56)** and **HiWi_P_norm** (-**0.60**). Moreover it can be observed that most of the other variables associated with the deformation of the tuffaceous rocks show a correlation (even if lower) with the temperature variables, while the other meteorological variables displays no correlation at all with the tuff deformation parameters (with the exception of a slight correlation with barometric pressure).

We further investigated the relationship between the deformation and the temperature variables. Therefore, we have computed the correlation at different lags in order to evaluate the effect of the temperature over time. In particular, we show in Table 12, the correlation among all the deformation variables and the air temperatures "**Temp**" at different lags (from 7 to 63 days, i.e. 1 to 9 weeks of delay), where the lag is expressed in number of days prior to the measurement of

the deformation. It can be noted that the variables **F04F1**, **F19I3-X** and **F19I3-Y** show a positive correlation with the air temperature almost at all different lags. Therefore, we suggest that there is a long-delayed time effect among them. Moreover, when we observe the different lags, a correlation between the **F05F2** and **F16F2** variables and the air temperature becomes evident. Overall, the correlation resulted increased within lags of 14-35 days.

Based on the observation on the daily sample, we decided to test the dependency of the deformation from the two

meteorological factors: temperature and wind pressure. The correlations between the variables measuring temperature and the variables measuring the wind pressure show negative and very low values (**e.g. -0-054** between **Temp** and **Wi-P_norm** and **-0.08,** between **Temp** and **HiWi-P_norm**). Obviously, all variables measuring both temperature and the wind effects are internally consistent and highly self-correlated. Therefore, we can perform a regression model where the dependent variable is the deformation measured by the variable **F04F1** and the explicative variables are **Temp** and

**Wi-P_norm.**

The model shows (Table 13) an adjusted R square of 0.63, so the linear model displays a high goodness-of-fit. Furthermore, the model is statistically significant (as p-values show). In Table 14 we report the estimates of model coefficients. Both the coefficients are significant as shown by the p values. We also observe a positive effect of both variables on the deformation at a daily level. Therefore, we infer that the deformation increases if temperature and wind

effect increase and the variation is defined by the values of the two-regression coefficients.

## 7 Discussion

Rock deformations involving five tuff blocks have been monitored for about 4 years (Dec. 2014 - Oct. 2018) along the Coroglio coastal cliff, together with local meteorological parameters, whose measurements began in January 2014. The selected unstable tuff blocks, characterized by a volume of 4 - 15 m$^3$, can be affected by toppling, planar and wedge

sliding failure kinematics. The used sensors (9 mono-axial crackmeters and 2 biaxial tiltmeters) captured several signs of tuff deformation through time as measured expansion and contraction of the fracture sheets and plane rotations above their accuracy (Tab. 2). Near-rock-surface air temperature (used as a proxy for surface rock temperature) and air temperature (together with other meteorological parameters) were measured respectively by three thermistors on the cliff and a weather station installed at a distance of ~ 1 km from the cliff. The whole dataset collected during the five



years of monitoring activity has been described; the data distribution (Figs. 6 - 12) and the temporal trends of the different parameters (Figs. 13 - 24) have been analyzed.

Based on a multivariate statistical analysis, we have recognized the relations between daily average values of tuff deformations and meteorological variables. Several positive correlations exist among rock deformation parameters and

temperature data (Tab. 12). Positive correlations increase if we introduce a time lag of 2 to 5 weeks for temperature variables (Tab. 13), suggesting that there is a delayed time effect between thermal oscillations and rock deformations. In some cases, the tuff deformation is negatively correlated to wind pressure intensity acting on the cliff. The regression analysis shows that if temperature and wind effect increase of 1 unit, deformation increases of 0.0114 and 0.0008, respectively. The coefficients of the model are significantly different from zero as the values of p-value shows.

The detected cyclic changes in opening of fractures and rock face rotations appear linked to seasonal and annual temperature variations (Fig. 25). Crack aperture data reveal a deformation pattern characterized by seasonal cyclical deformation trends with strong temporal temperature coupling. In detail, when maximum temperatures level out (July-August), deformation continues to increase up to 30-40 days and reaches the maximum values during late August-September, each year. Minimum deformation values usually occur during winter - early spring (from January to April),

after the seasonal cooling period. Some crackmeters show secondary peaks during autumn-winter period, uncorrelated to temperature variation.

Similarly, we found comparatively little influence between rain and wind with the fracture deformation (Figs 26-28). The rain seems to cause only sudden decreases in fracture opening during intense rainstorm (see events around 27/01/2015, 27/10/2015, 27/08/2017 and 27/09/2017, Fig. 26). On the contrary, humidity and barometric pressure seem

to exert no influence on fracture deformation (Figs 29-30).

The temperature exerts the dominant role in driving daily average values of deformation, as the fracture deforms both synchronously and in a delayed way with temperature. Deformation of rocks measured by crackmeter sensors are partly linked to the bulk volume variation of the tuff, as a response to temperature changes. Tiltmeters measurement are also strongly forced by temperature annual cycles showing a cumulate increase from 2015 to 2018 (Fig. 31).

The direct result of heating rock is expansion, a process well studied and quantified in several environmental conditions (Chau and Shao, 2006; Collins and Stock, 2016; Eppes, 2016; Richter and Simmons, 1974). Cumulative fracture growth and opening can occur under typical present-day meteorological conditions in many settings (Eppes and Keanini, 2017; Lamp et al., 2017). Several diurnal and seasonal cycles of heating and cooling may lead to deformation and crack opening and propagation. Both increasing temperature and temperature fluctuations may also produce and enhance

fracturing.

On a seasonal cycle, cumulative positive deformation peaks occur during the warmest months (June throughout September) yielding values up to 4 mm higher than the maximum negative deformation occurring during the coolest



months (December through March; Fig. 25). This suggests that some rockfalls might be more likely during hot summer months, when crack sheets are at their maximum opening along the cliffs. Nevertheless fracture opening is known to be a nonlinear process (Lawn, 1993), so we cannot simply use our measured deformation rates to predict future detachments of fracture sheets.

Our results have important implications for the triggering of rockfalls in cliffed landscapes. In fact, our measurements indicate that seemingly static tuffaceous landscapes are actually dynamic slopes. Along the Coroglio cliff tuff blocks can deform in and out of a near-vertical cliff face by up to 1 mm on daily basis and up to 4 mm on annual scale. These data demonstrate the inherent instability of studied tuff cliffs as well as other rocky cliffs worldwide (Collins and Stock, 2016), mainly during summertime (Ishikawa et al., 2004; Gunzburger et al., 2005; Hasler et al., 2012; Vargas et al.,

2012; Stock et al., 2013; Collins & Stock, 2016).

## 8 Data availability

The databases presented and discussed in this article are available for downloading from PANGAEA Repository. Deformation and meteorological datasets are respectively provided in separated files in tab format at https://doi.pangaea.de/10.1594/PANGAEA.896000 (Matano et al., 2018) (deformation data with frequency sampling of

30 minutes) and at https://doi.pangaea.de/10.1594/PANGAEA.899562 (Fortelli et al., 2019) (meteorological data with frequency sampling of 10 minutes).

## 9 Conclusions

The Coroglio cliff shares broad characteristics with several tuff cliff from the Campi Flegrei coastal area, and elsewhere in the world. The Coroglio CC-MoSys monitoring system, installed in December 2014, captured some of the tuff

deformation behavior under the different meteorological conditions occurred during about 5 years.

The data recorded by the Coroglio MoSys have contributed to our understanding of temperature-tuff deformation relationships in the analyzed cliff. Micro-deformation of rocks measured by geotechnical sensors reveals a general cyclic trend, possibly linked to the bulk volume variation of the rocks, as a response to seasonal and daily temperature variations. This research provides a first contribute to the understanding of the rates of geomorphic evolution of coastal

tuff cliff and its relationships with forcing factors (e.g. meteo-marine weathering, human actions, volcano-tectonic activity) in the perspective of early-warning actions and policies.



**Author contribution**

Data collection was done by FM, MC, AF and MSa. Analysis of the data was done by FM and AF. Statistical analysis was done by GS and MSp. The initial draft of the paper was written by FM with contributions by AF (sect. 5), GS and MSp (sect. 6). All authors contributed to the final version of the text.

5   **Competing interests**

The authors declare that they have no conflict of interest.

**Acknowledgements**

Financial support for this research was provided by the Programma Operativo Nazionale (PON) funded by the Italian Ministry of University and Research-Project PON-MONICA (grant PON01_01525).

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



**Figures**



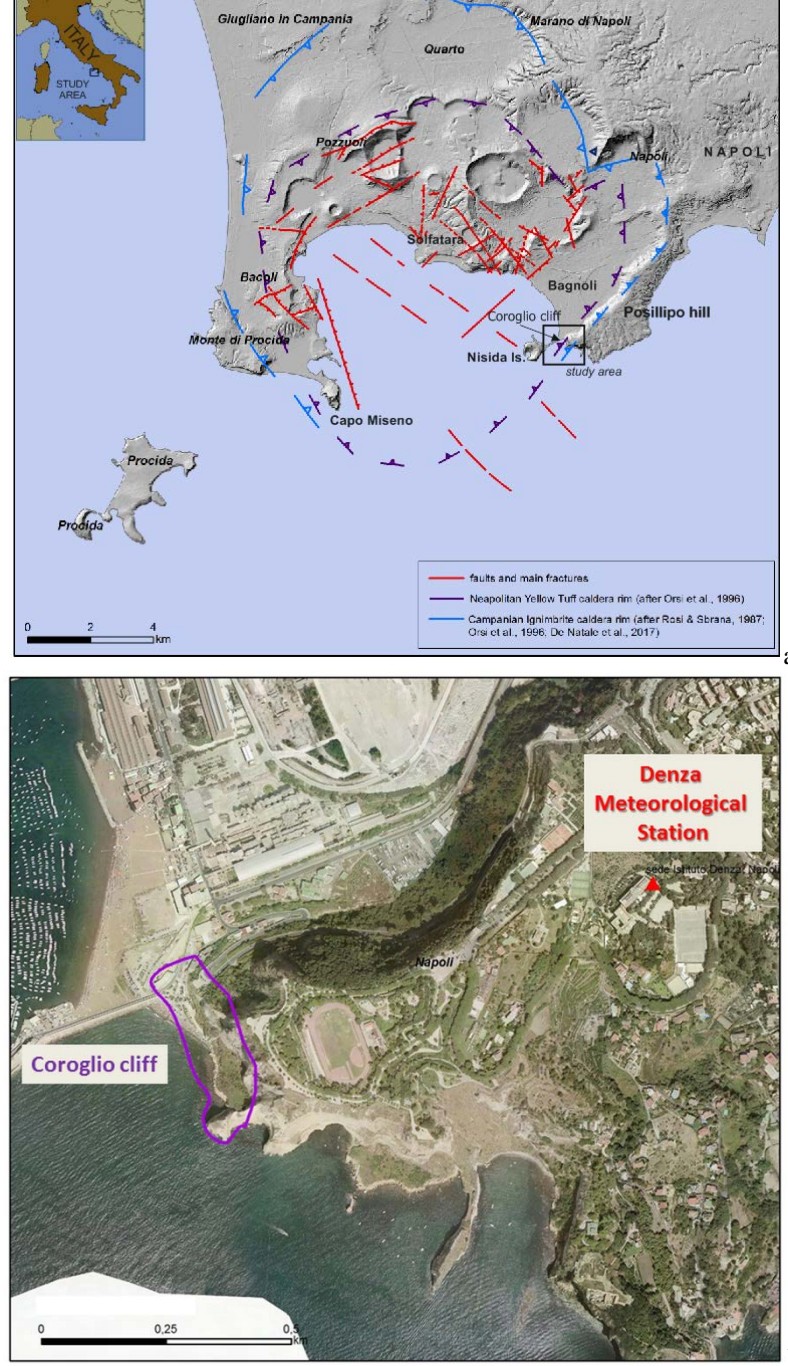

**Figure 1: a - Study area location and Campi Flegrei area. b – Study area detailed orthophoto (2004).**

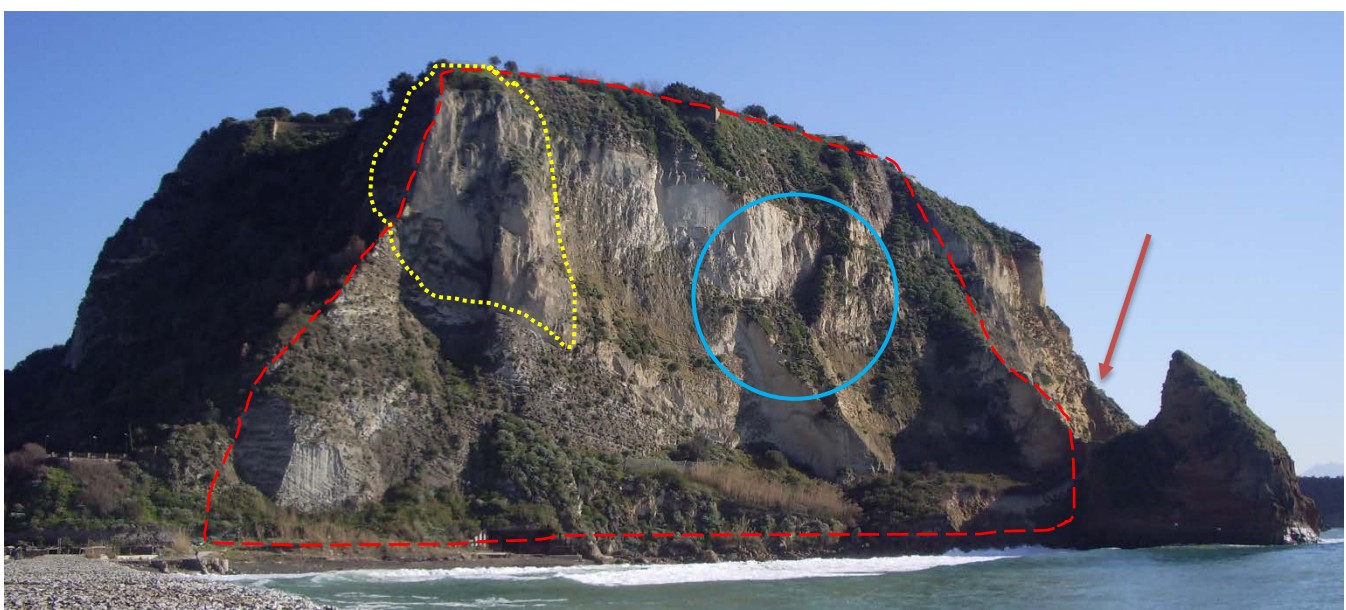

**Figure 2: Coroglio cliff view from WSW. Red dashed line shows the area mapped in Fig. 3; yellow dotted line shows the area involved by reinforcement works; light blue circle shows the area with unstable tuff blocks; red arrow shows the detachment area of the failure occurred around 1990, located outside the area mapped in Fig.3.**

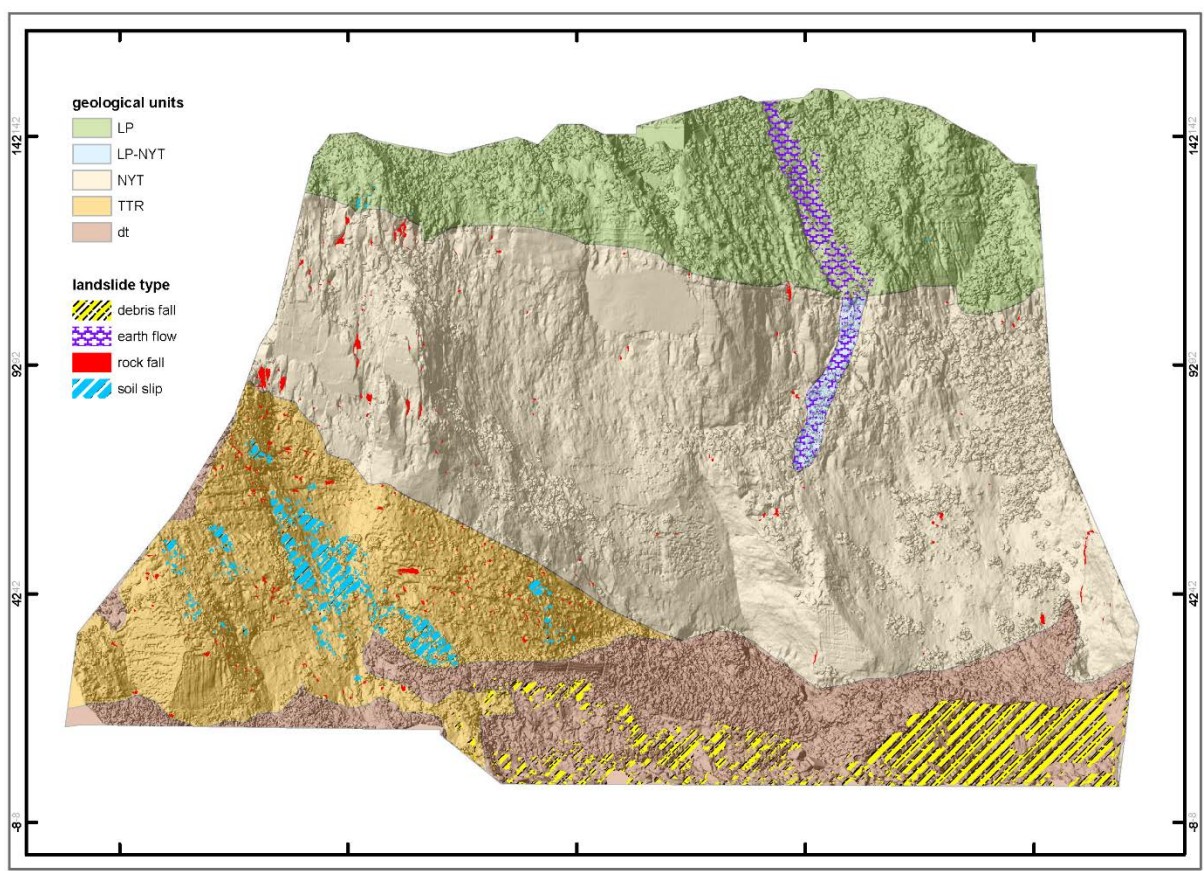

**Figure 3: Geological vertical map of the Coroglio cliff (modified after Matano et al., 2016). Landslide types occurred during 2013-2015 interval (Caputo et al., 2018) are also showed. Geological units: LP, stiff to loose recent pyroclastic deposits and soils; LP-NYT, thin layer of LP deposits on NYT unit; NYT, Neapolitan Yellow Tuff; TTR, Trentaremi Tuff; dt, slope talus breccia and gravelly beach deposits.**



**Figure 4: Geostructural vertical map (modified by Matano et al., 2016). Black boxes show location of the monitored unstable tuff blocks. Set legend (dip/dip direction): F1a (>65°/30-50° and 220-235°), F1b (>65°/50-60° and 235-245°), F1c (>65°/55-65° and 245-255°), F2 (>65°/0-30° and 180-220°), F3 (>70°/60-110° and 255-280°), F4 (>70°/110-180° and 300-355°), F5 (20-65°/50-195°), F6 (< 20°/0-360°)**



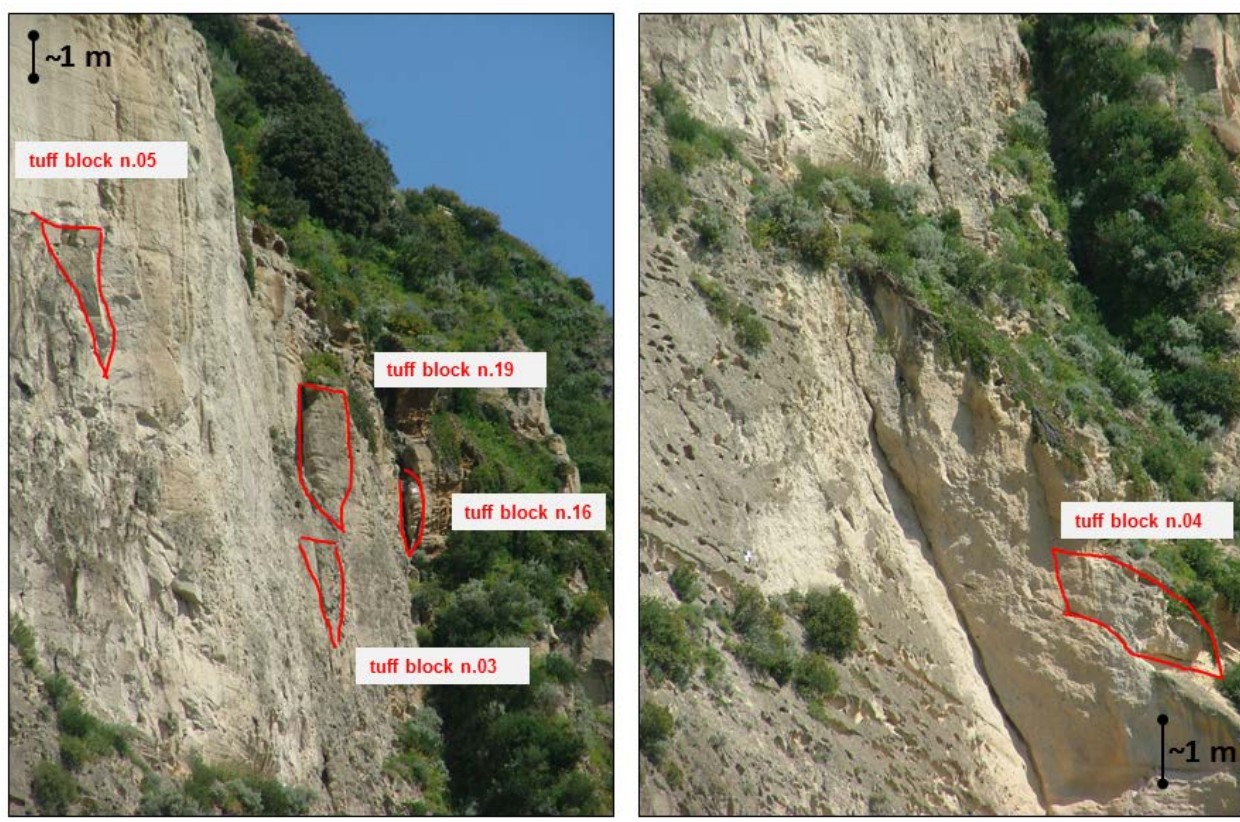

**Figure 5: Unstable tuff blocks of the NYT unit selected for monitoring activity (red lines).**

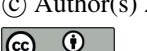



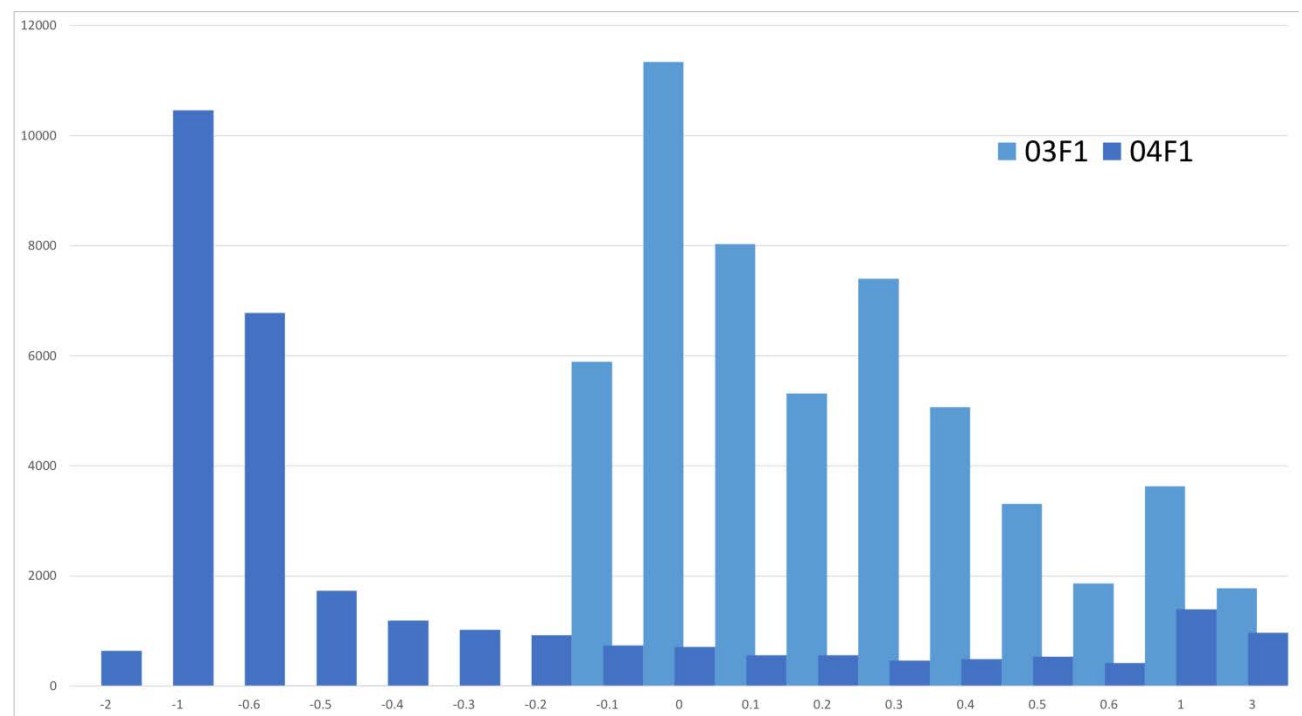

**Figure 6: Data distribution of sensors 03F1 and 04F1. The number of measurements is reported on y axis, and the value of deformation (in mm) on x axis.**



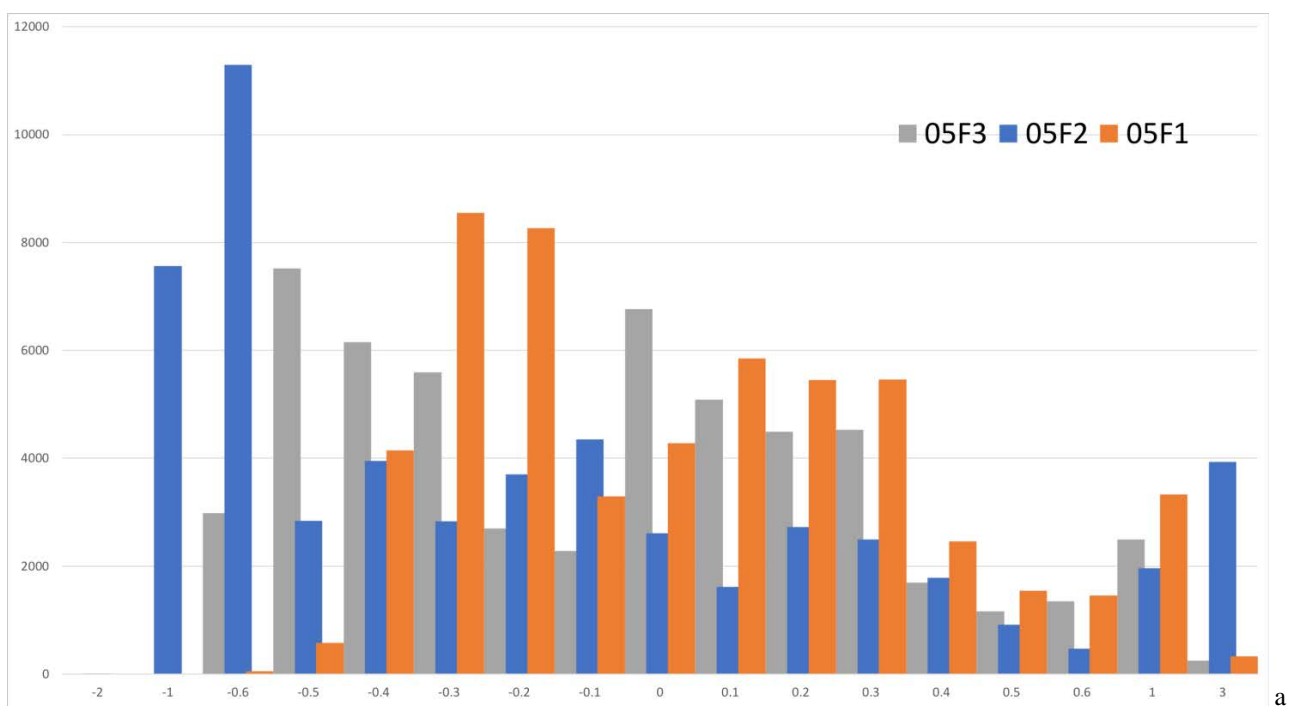

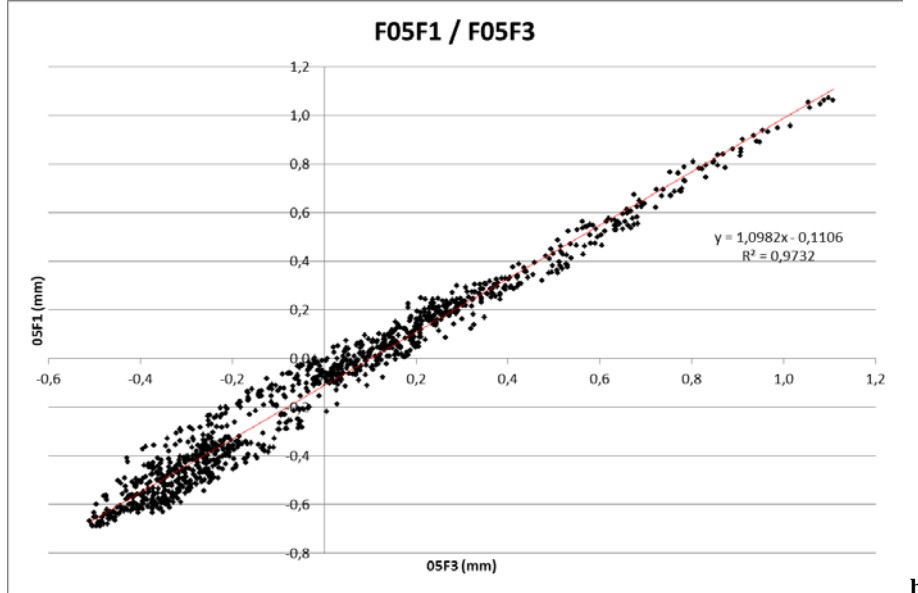

**Figure 7: a - Data distribution of sensors 05F1, 05F2 and 05F3. The number of measurements is reported on y axis, and the value of deformation (in mm) on x axis. b – Correlation plot between 05F1 and 05F3 data.**

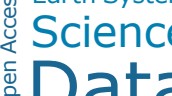



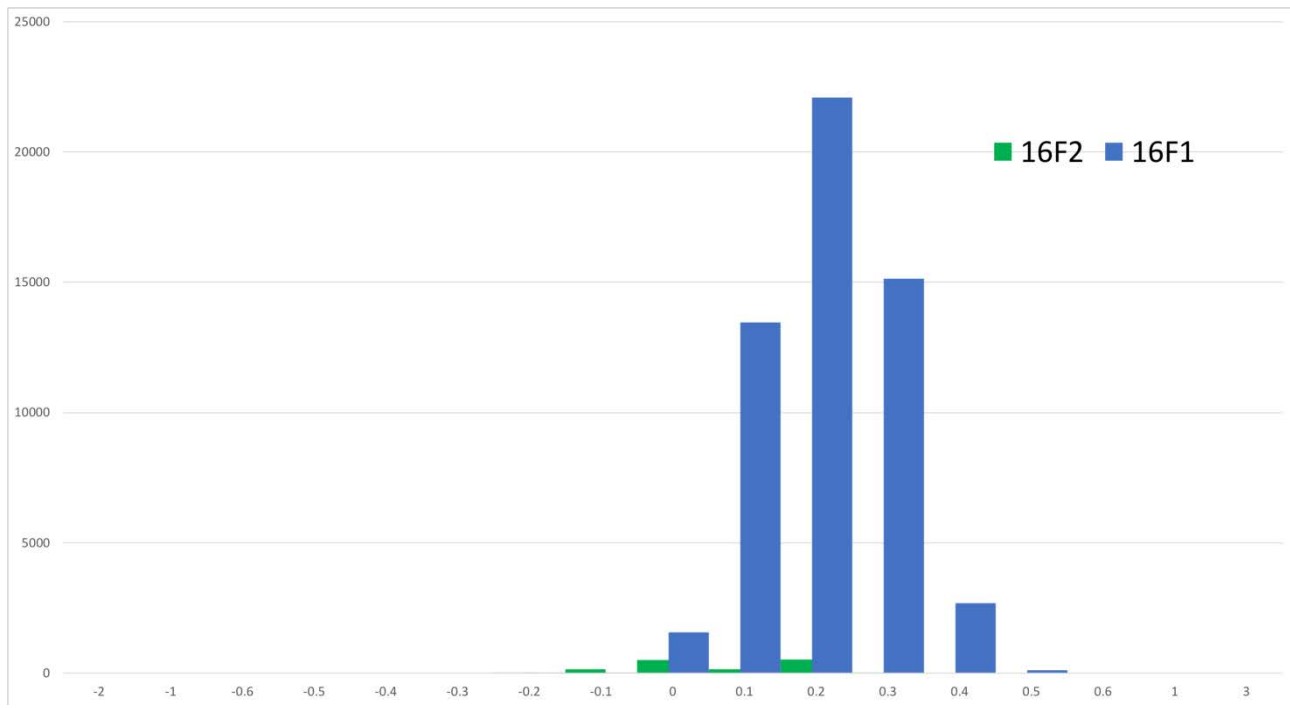

**Figure 8: Data distribution of sensors 16F1 and 16F2. The number of measurements is reported on y axis, and the value of deformation (in mm) on x axis.**



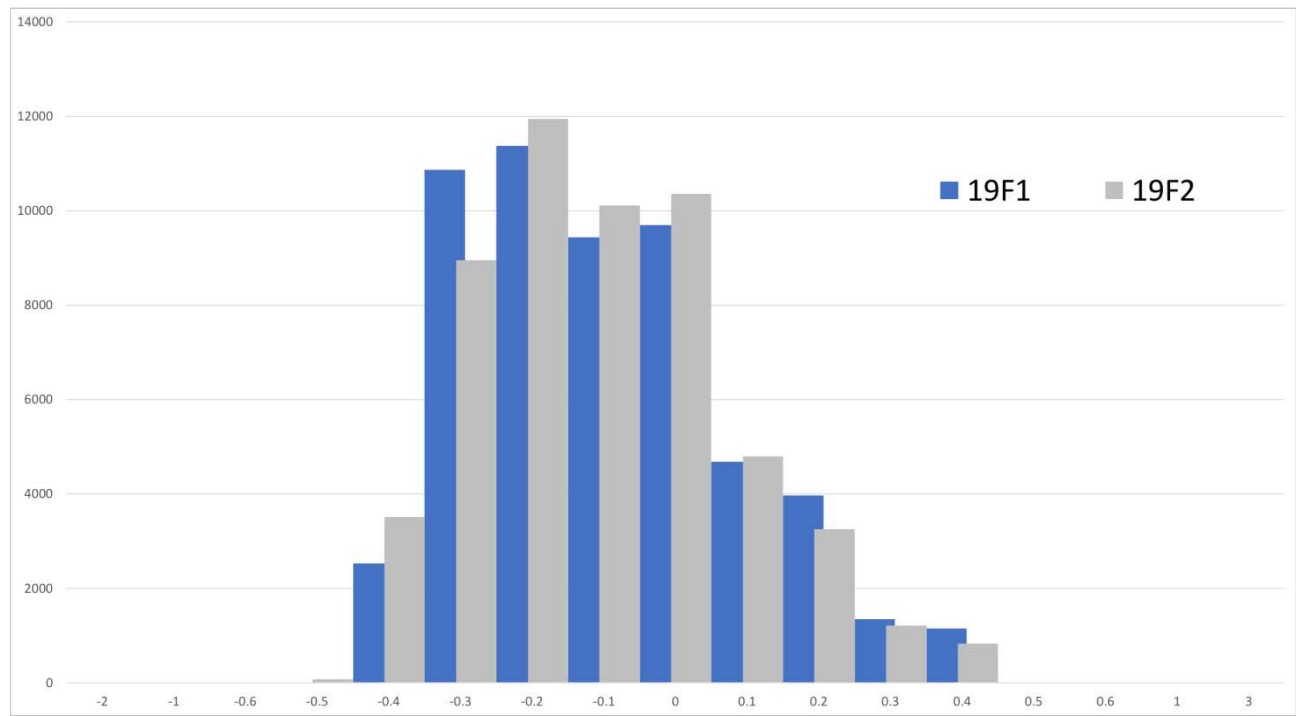

**Figure 9: Data distribution of sensors 19F1 and 19F2. The number of measurements is reported on y axis, and the value of deformation (in mm) on x axis.**





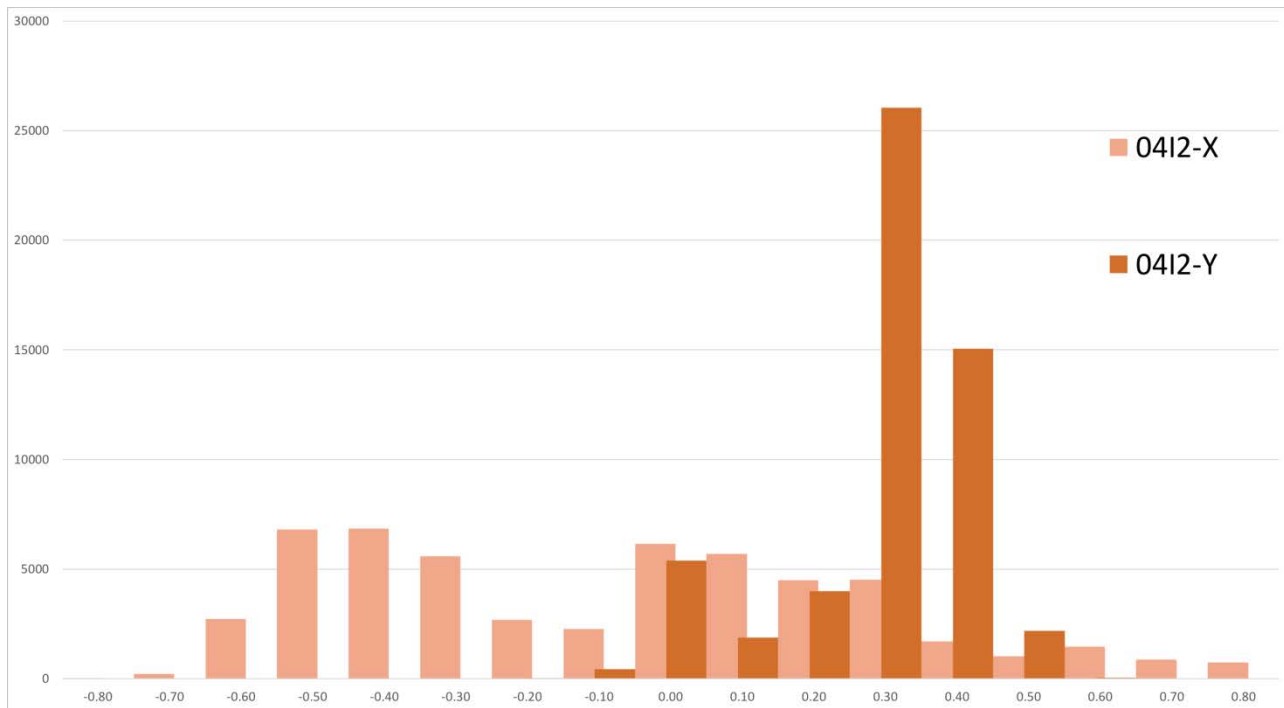

**Figure 10: Data distribution of sensors 04I2-X and 04I2-Y. The number of measurements is reported on y axis, and the value of deformation (in degree) on x axis.**





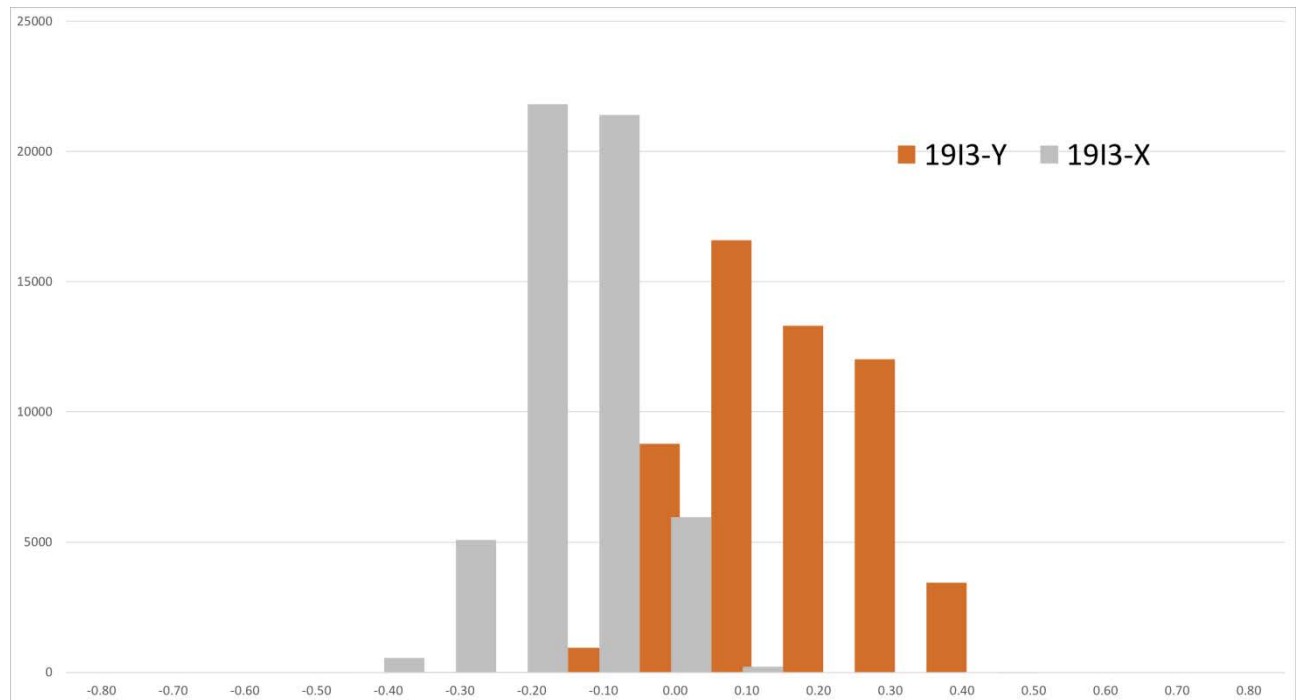

**Figure 11: Data distribution of sensors 19I3-X and 19I3-Y. The number of measurements is reported on y axis, and the value of deformation (in degree) on x axis.**



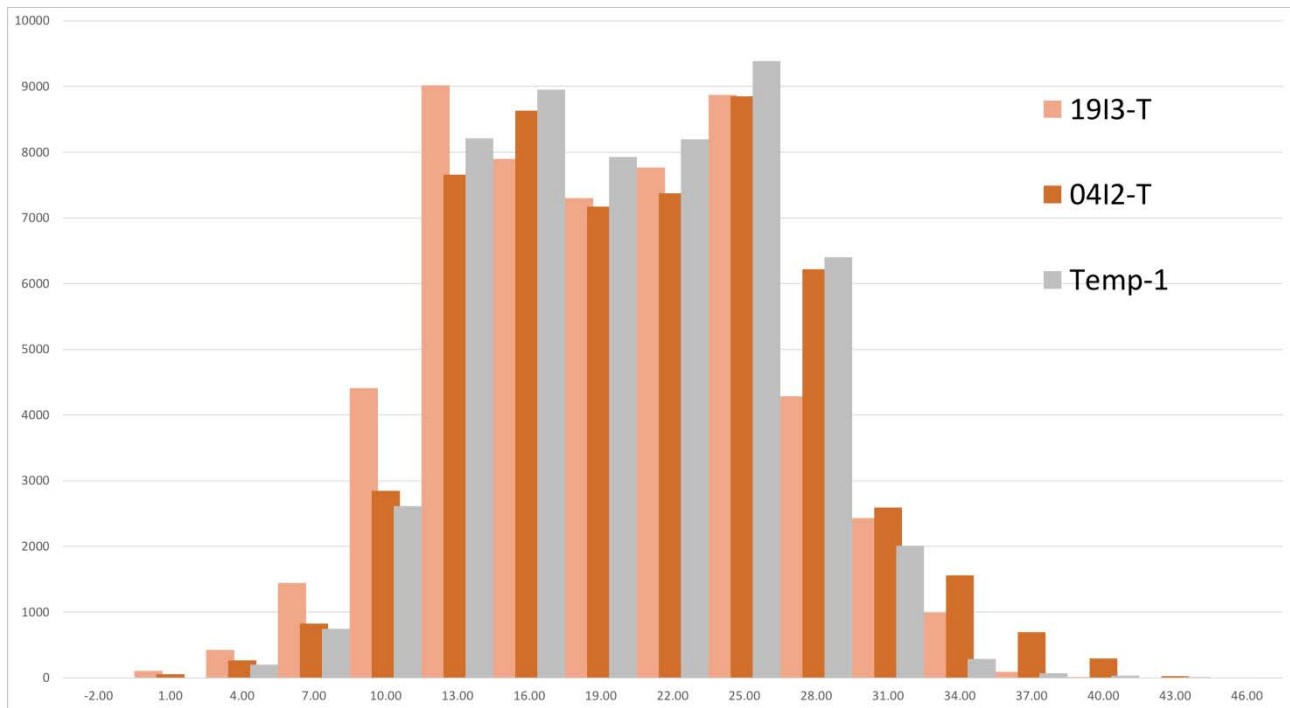

**Figure 12: Data distribution of thermistors 19I3-T, 04I2-T and Temp-1. The number of measurements is reported on y axis, and the value of deformation (in degree Celsius) on x axis.**



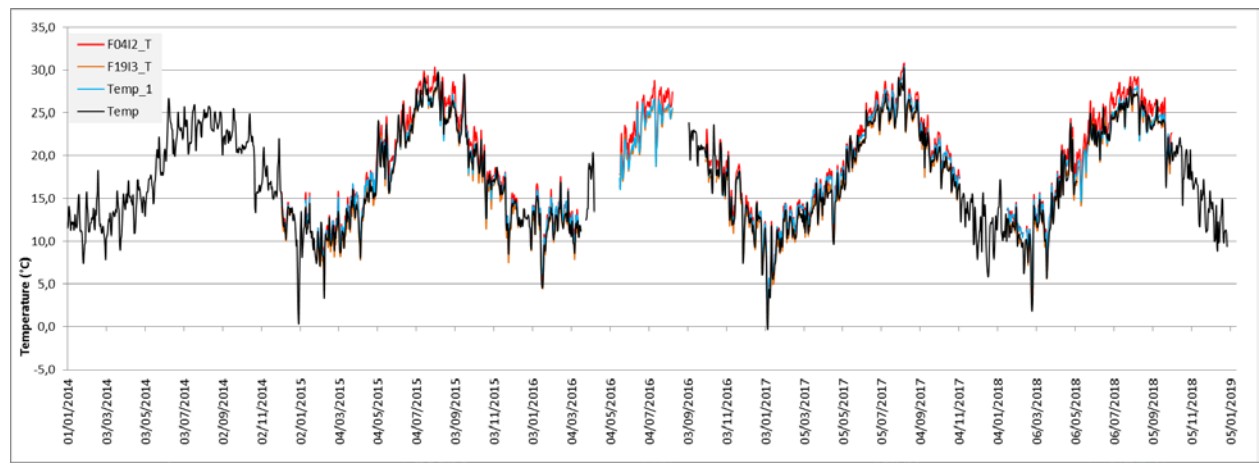

**Figure 13: Temperature (daily average data) time series showing seasonal and annual variation patterns. Long-term trends, based on about five consecutive years of data, show annual cyclicity. Air temperature and near-rock-surface air temperature are fully synchronous; peaks in near-rock-surface air temperatures (19I3-T, 04I2-T and Temp-1) are sometimes higher than those of air temperature (Temp). Data are incomplete in some months.**







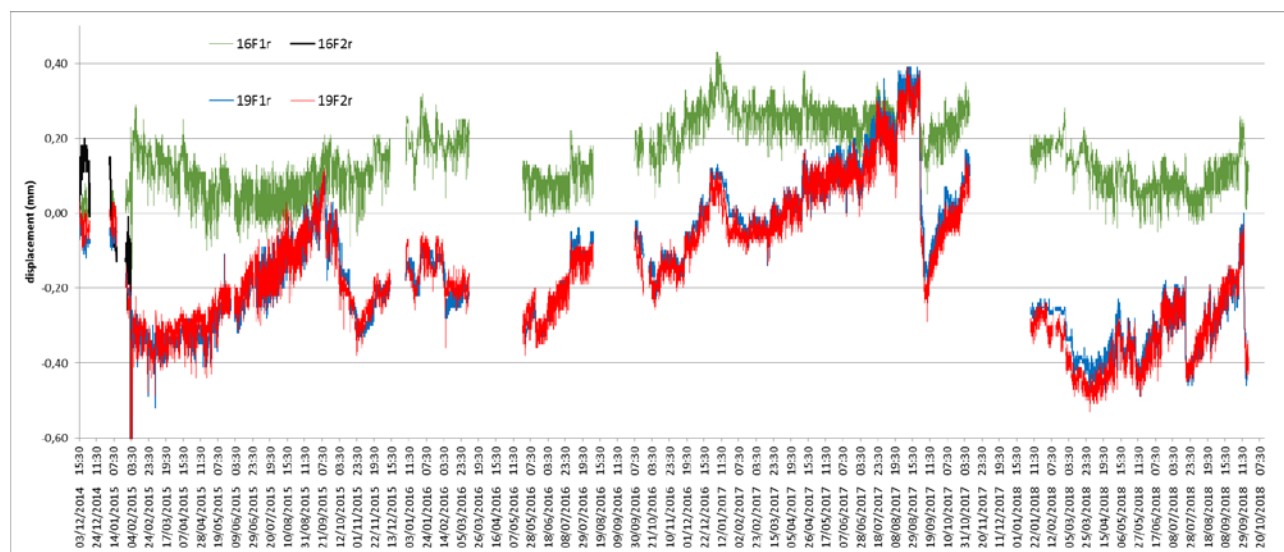

**Figure 14: Crack aperture time series data showing seasonal and annual deformation patterns. Long-term trends, based on about four consecutive years of data (December 2014 to October 2018), show from +0,1 to +1,2 mm cumulative deformation. Data are incomplete in some months.**



**Figure 15: Angle variation time series data showing seasonal and annual deformation patterns. Long-term trends, based on about four consecutive years of data (December 2014 to October 2018), show from 0,1° to 0,4° cumulative angle variation. Horizontal and vertical angle variation show opposite evidences for each block. Data are incomplete in some months.**



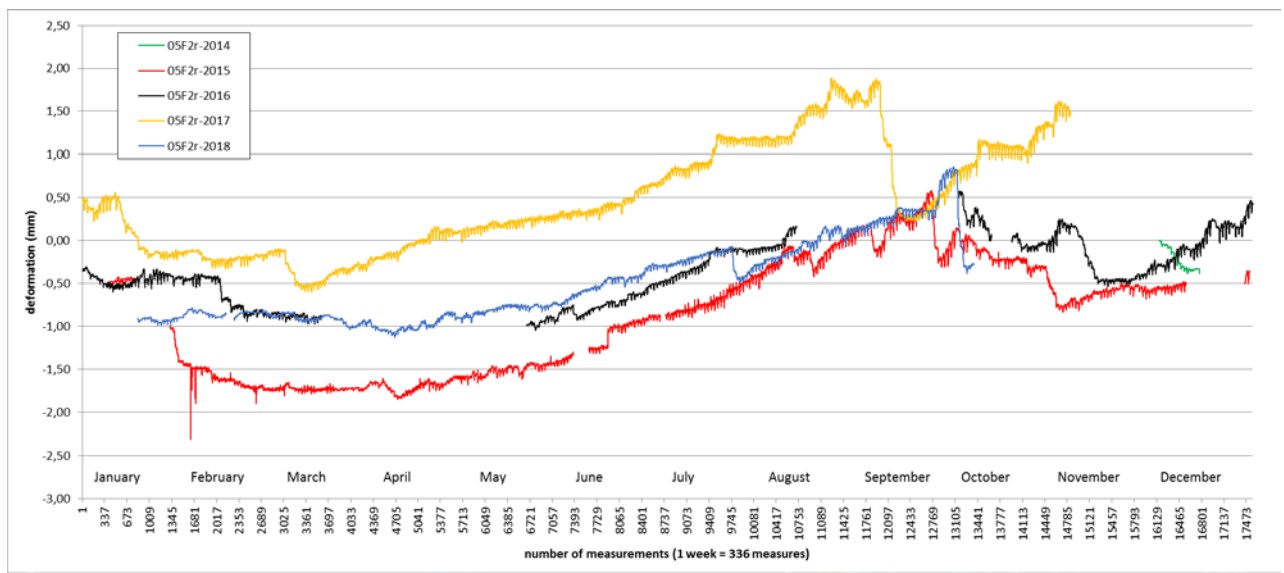

**Figure 16: Annual time series plots of 05F2 crackmeter showing partly repetitive seasonal deformation patterns.**





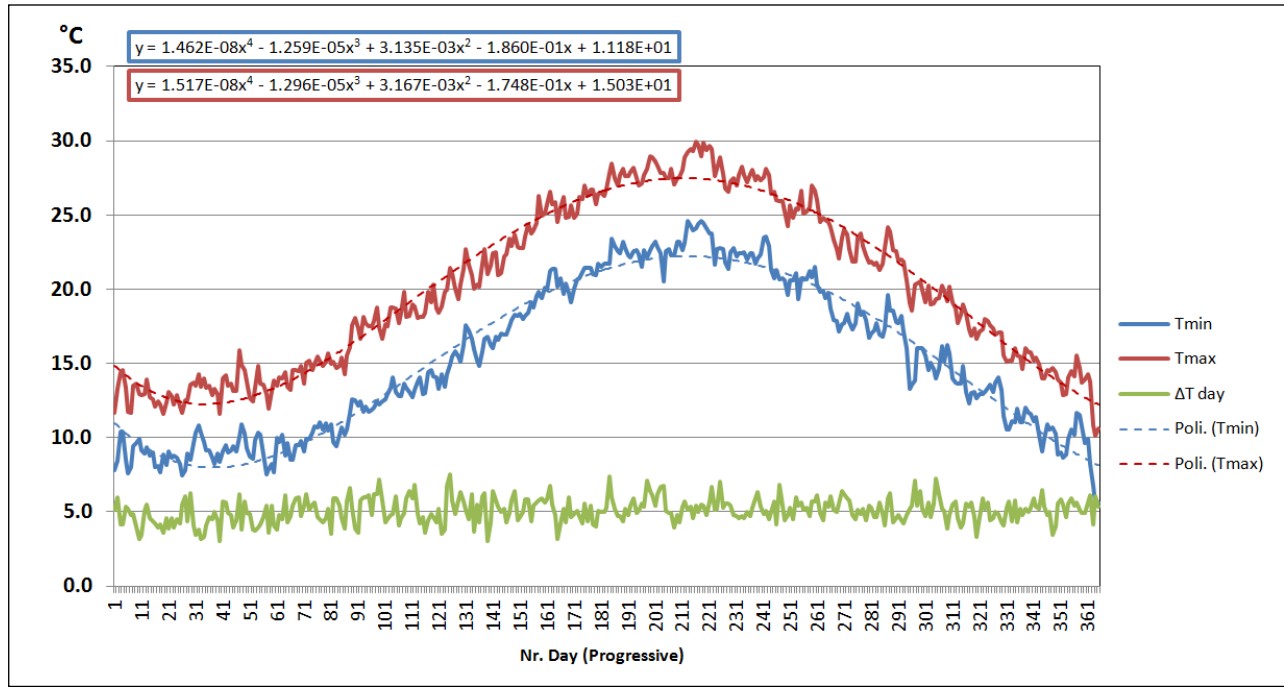

**Figure 17: Average daily thermal trend, referred to 2014-2018, recorded by Denza weather station with 4ᵗʰ grade polynomial equation. Maximum temperature values are indicated with red line, minimum values with blue line and daily temperature range with green line.**





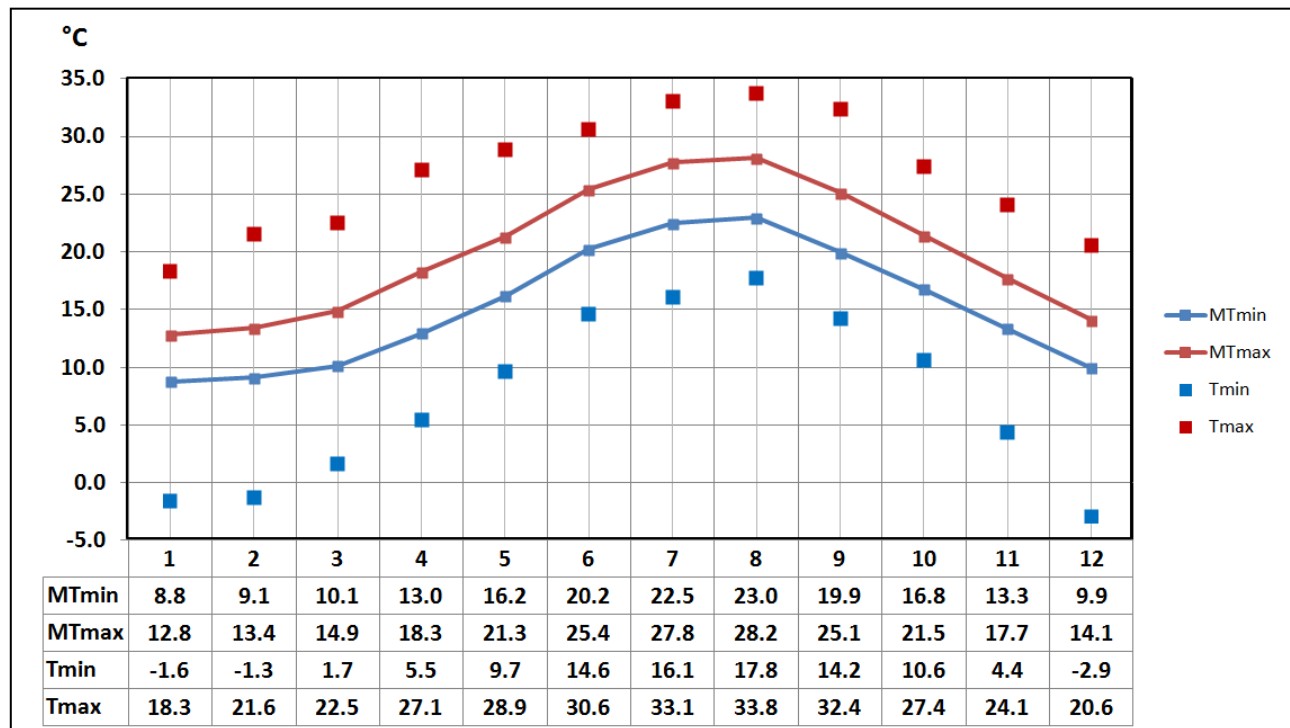

| °C | 1 | 2 | 3 | 4 | 5 | 6 | 7 | 8 | 9 | 10 | 11 | 12 |
|---|---|---|---|---|---|---|---|---|---|---|---|---|
| MTmin | 8.8 | 9.1 | 10.1 | 13.0 | 16.2 | 20.2 | 22.5 | 23.0 | 19.9 | 16.8 | 13.3 | 9.9 |
| MTmax | 12.8 | 13.4 | 14.9 | 18.3 | 21.3 | 25.4 | 27.8 | 28.2 | 25.1 | 21.5 | 17.7 | 14.1 |
| Tmin | -1.6 | -1.3 | 1.7 | 5.5 | 9.7 | 14.6 | 16.1 | 17.8 | 14.2 | 10.6 | 4.4 | -2.9 |
| Tmax | 18.3 | 21.6 | 22.5 | 27.1 | 28.9 | 30.6 | 33.1 | 33.8 | 32.4 | 27.4 | 24.1 | 20.6 |

**Figure 18: Average monthly thermal trend, referred to 2014-2018, recorded by Denza weather station. Maximum temperature values are shown with red line and dots, minimum values with blue line and dots.**



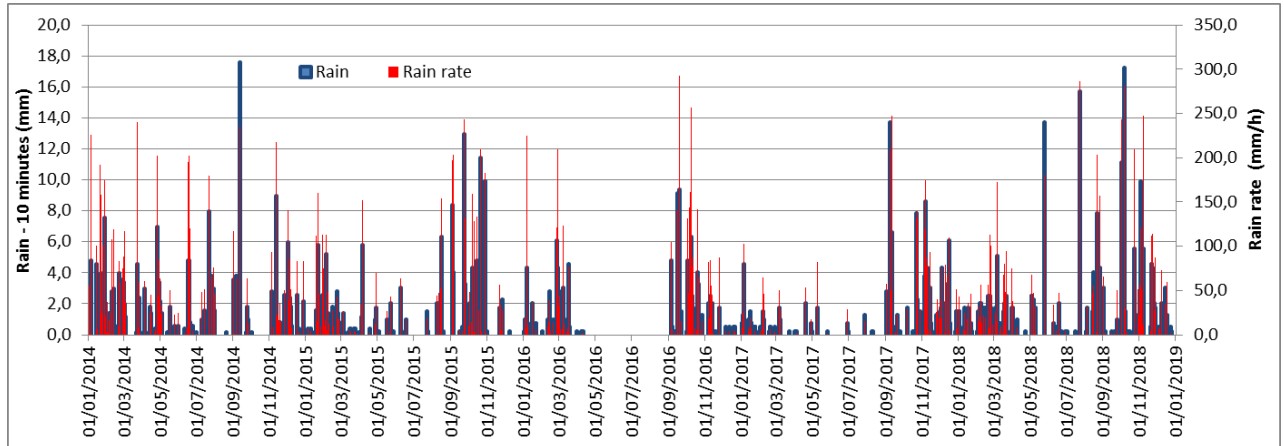

**Figure 19: Rain and rain rate time series data showing seasonal and annual variation patterns in the Denza station during 2014-2018.**

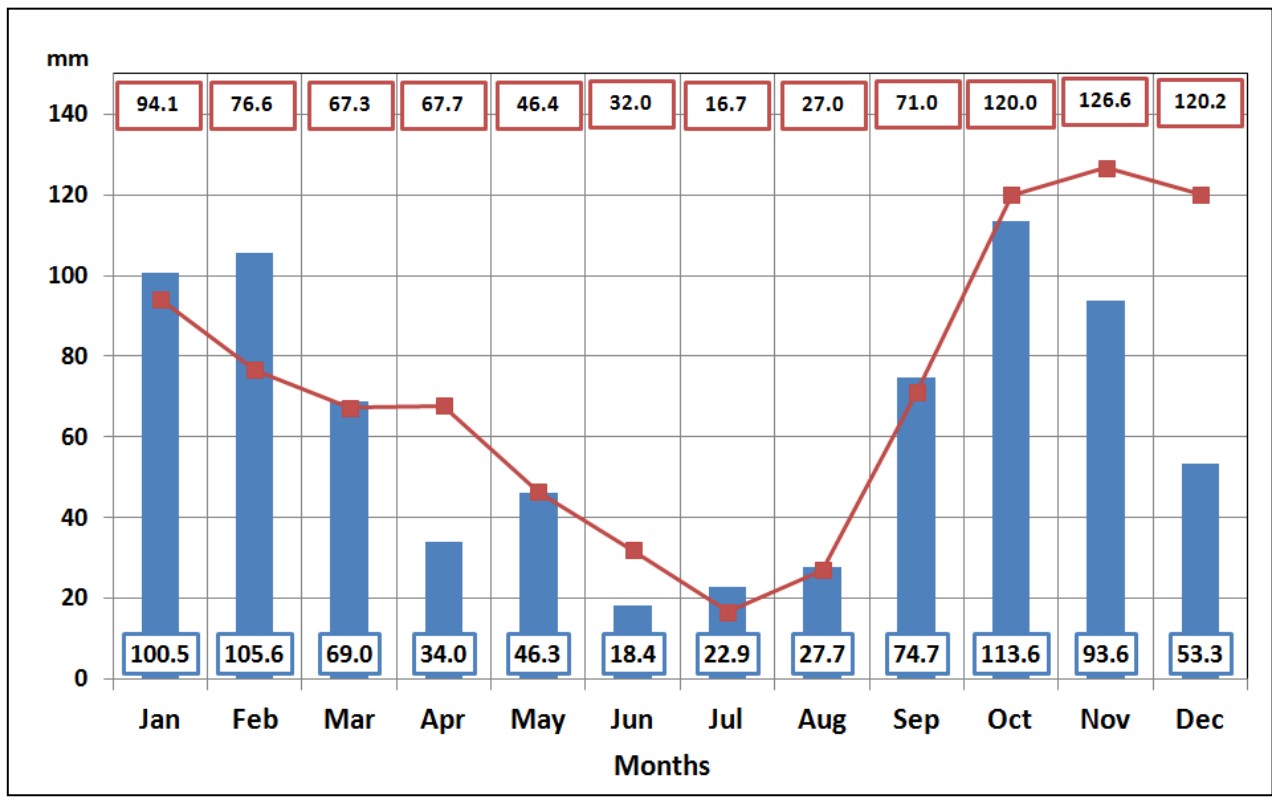

**Figure 20: Average cumulative monthly rainfall data histogram referred to 2014-2018. Diagram reports also average monthly values of rainfall, measured at Meteorological Observatory of University of Naples "Federico II" since 1872 (Mazzarella, 2006). Rainfall values are reported, respectively, in blue squared cells and red squared cells.**



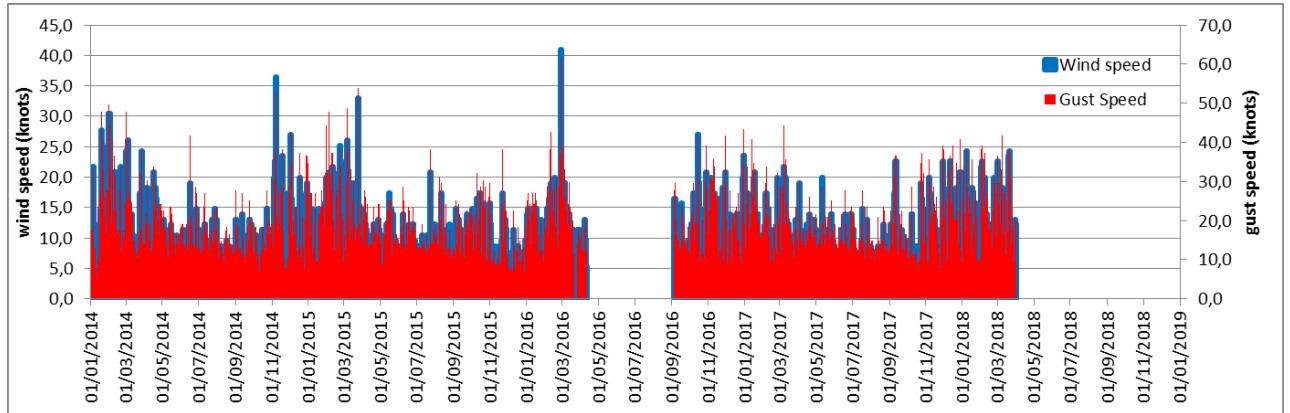

**Figure 21: Wind (Wi_speed) and gust (HiWi_spe) speed time series data showing seasonal and annual variation patterns in the Denza station during 2014-2018.**



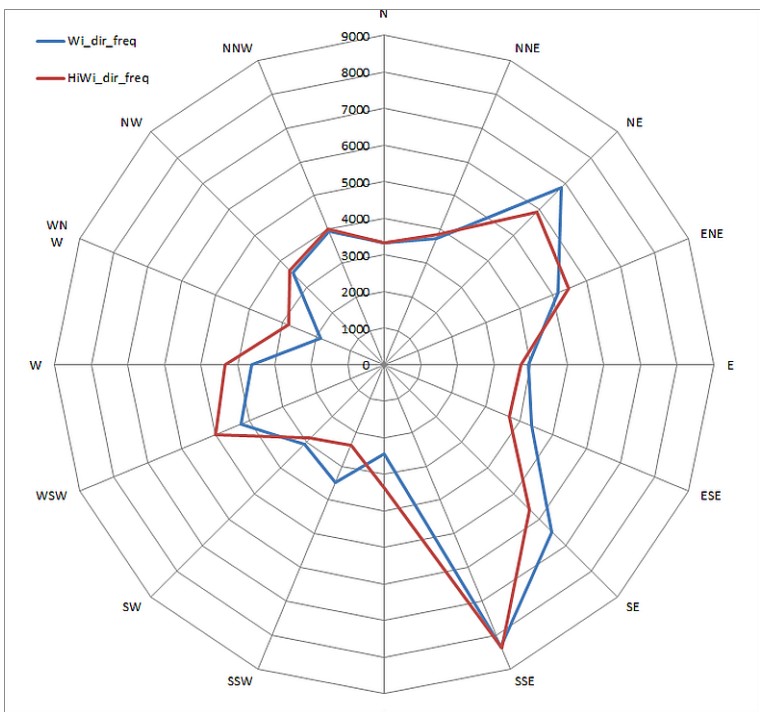

**Figure 22: Frequency distribution of Wind (Wi_dir) and gust (HiWi_dir) directions measured during 2014-2018.**





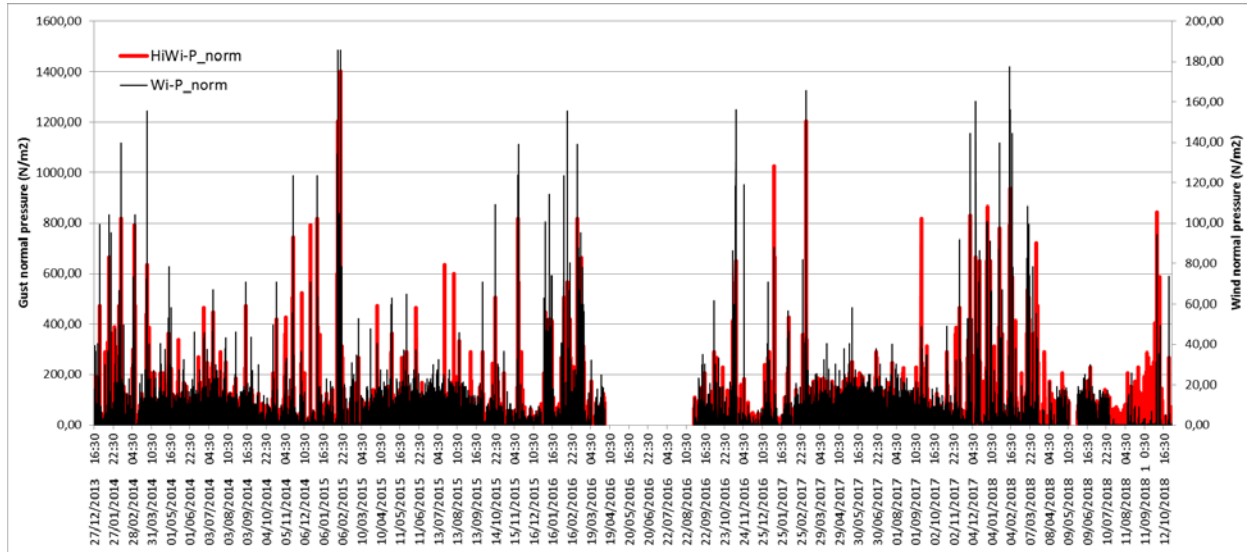

**Figure 23: Wind (Wi-P_norm) and gust (HiWi-P_norm) normal pressure time series data showing seasonal and annual variation patterns in the Denza station during 2014-2018.**



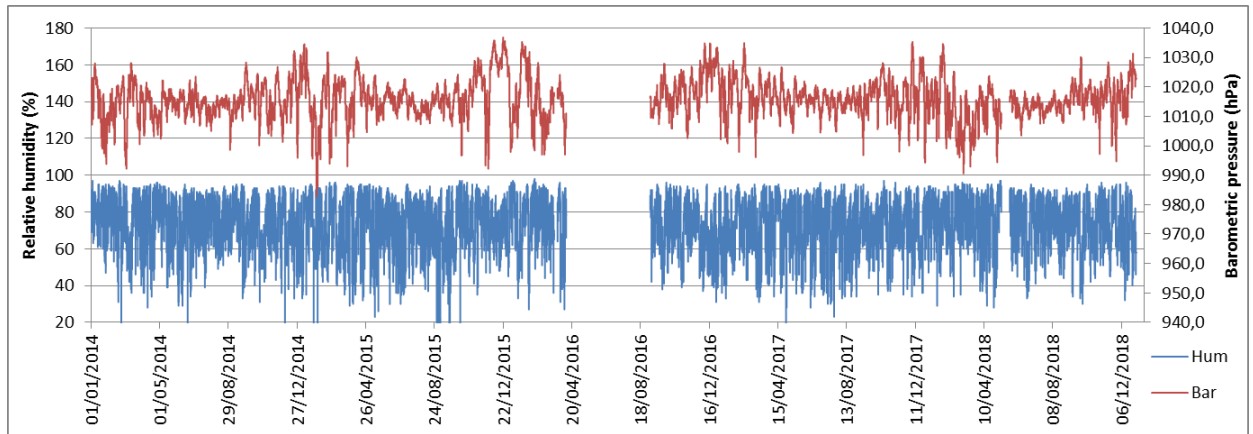

**Figure 24: Barometric pressure and humidity time series data showing seasonal and annual variation patterns in the Denza station during 2014-2018.**



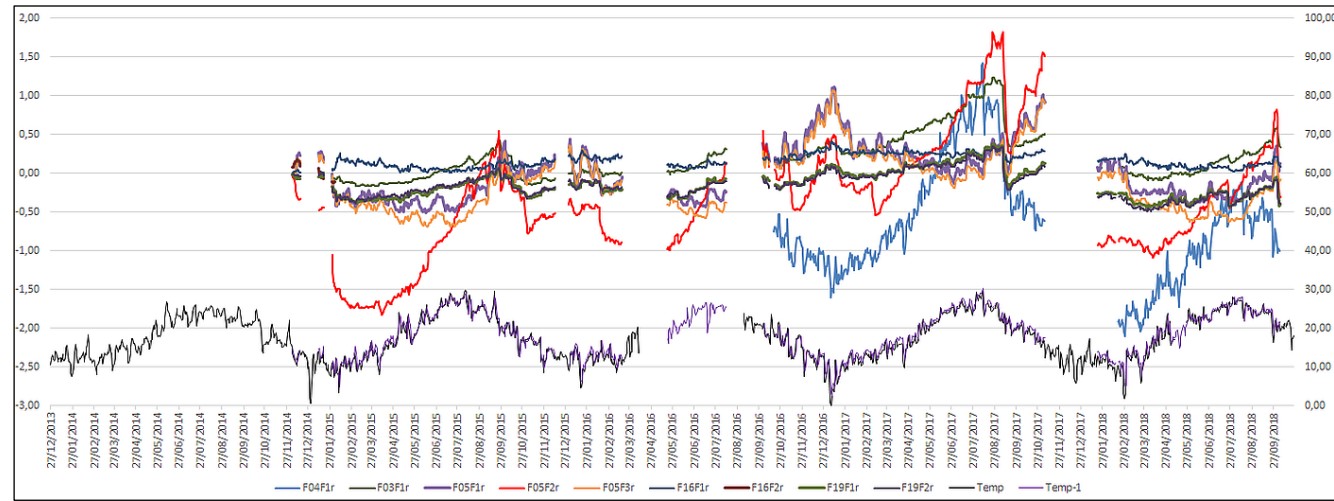

**Figure 25: Comparison between crackmeter measurements (expressed in mm, left vertical axis) and temperature (expressed in °C, right vertical axis) daily data time series.**

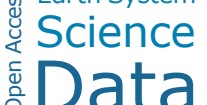

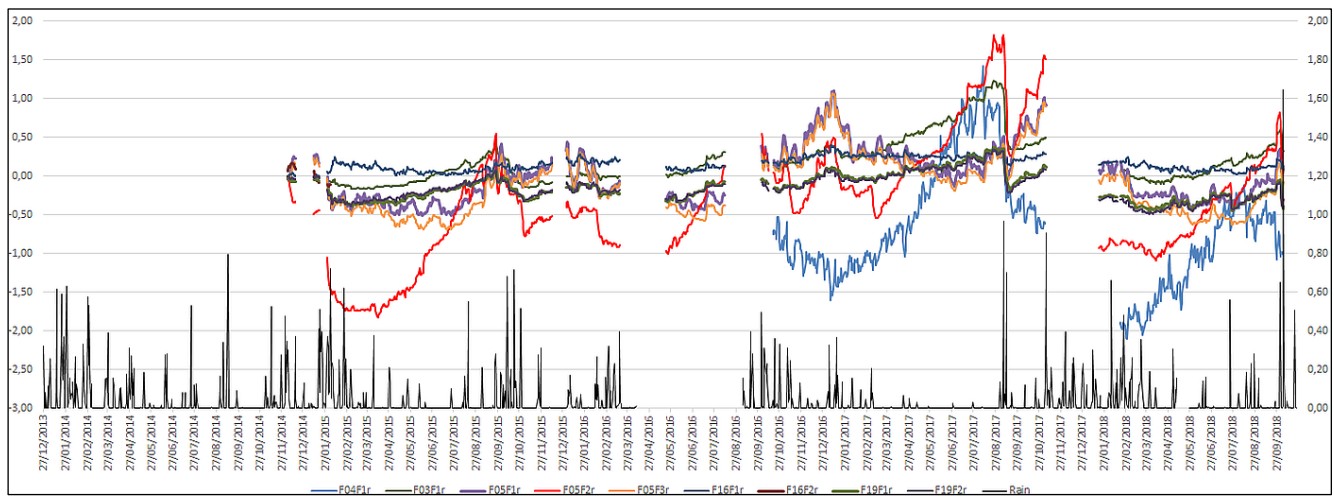

**Figure 26: Comparison between crackmeter measurements (expressed in mm, left vertical axis) and rainfall amount (expressed in mm, right vertical axis) daily data time series.**

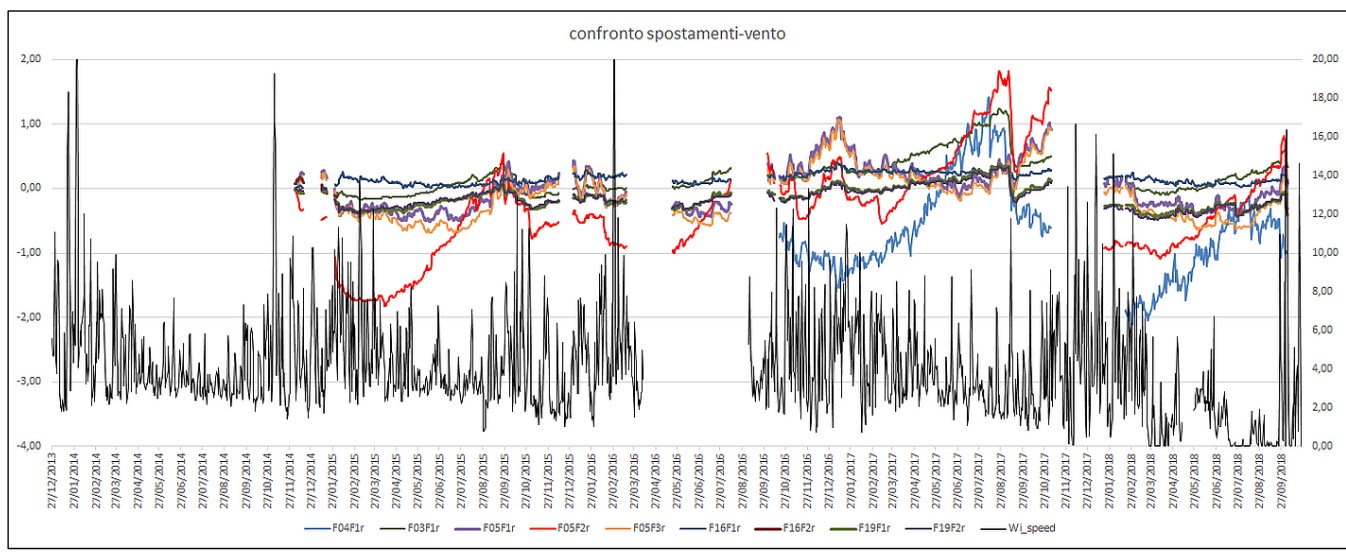

**Figure 27: Comparison between crackmeter measurements (expressed in mm, left vertical axis) and wind speed (expressed in knots, right vertical axis) daily data time series.**

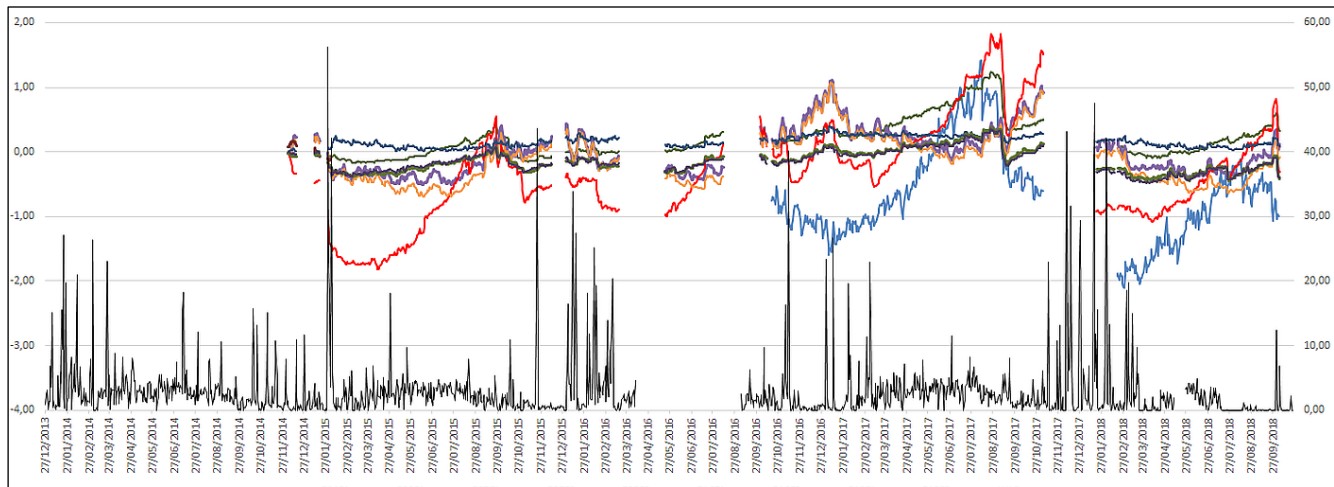

**Figure 28: Comparison between crackmeter measurements (expressed in mm, left vertical axis) and wind normal pressure (expressed in Nm⁻², right vertical axis) daily data time series.**



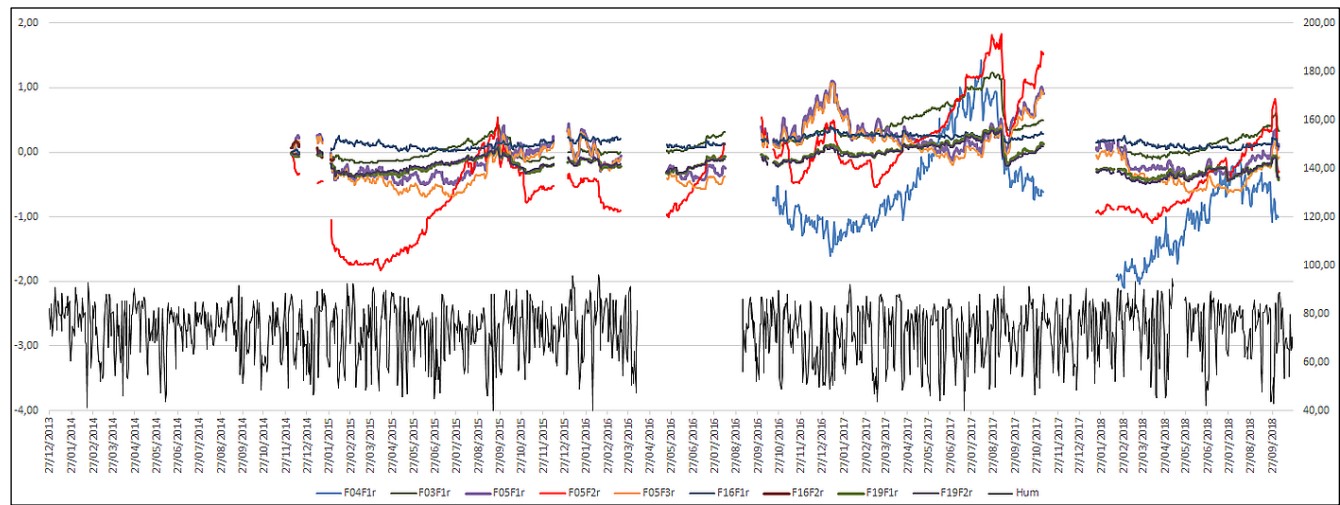

**Figure 29: Comparison between crackmeter measurements (expressed in mm, left vertical axis) and humidity (expressed in %, right vertical axis) daily data time series.**





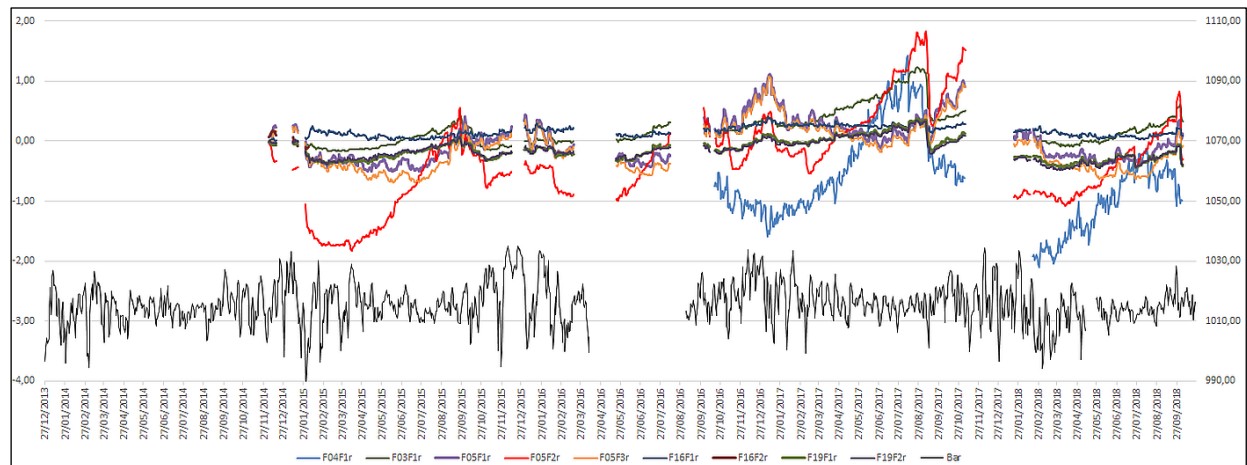

**Figure 30: Comparison between crackmeter measurements (expressed in mm, left vertical axis) and barometric pressure (expressed in hPa, right vertical axis) daily data time series.**

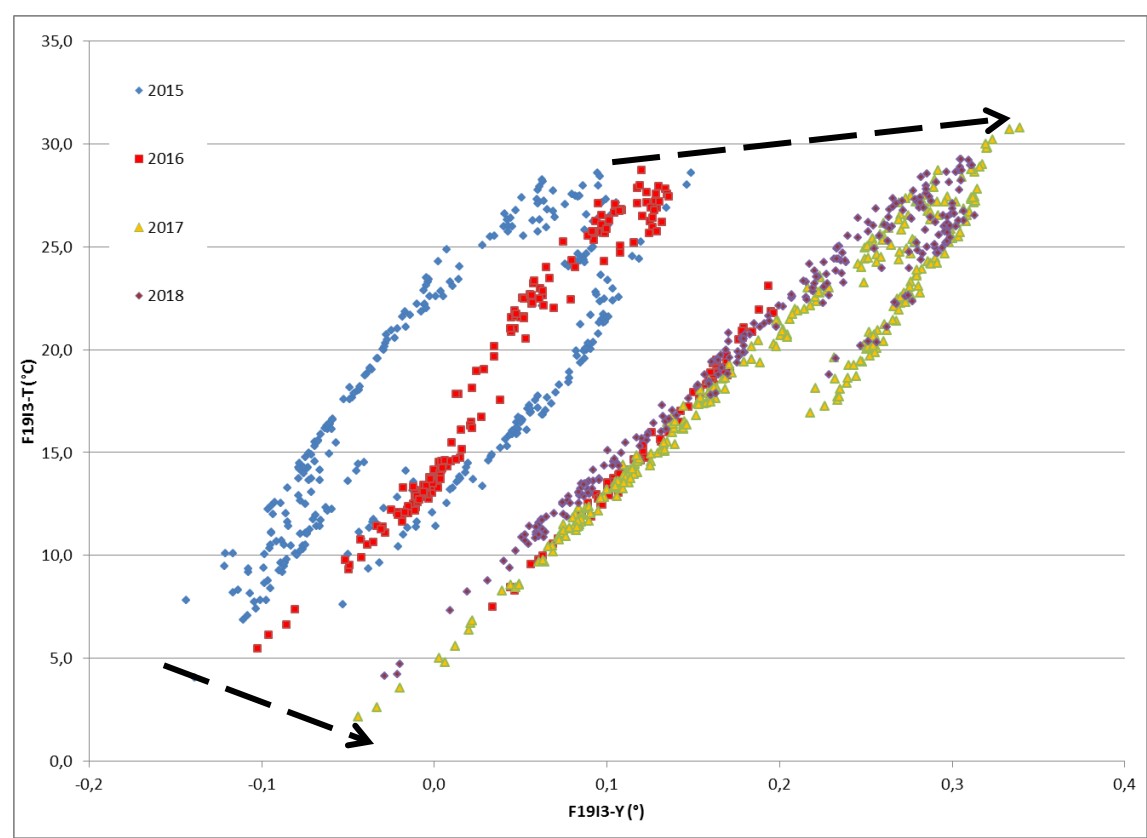

**Figure 31: Progressive increase of tiltmeter F19I3-Y measurements values (see dashed arrows) from 2015 to 2018 referred to local temperature.**



10        **Tables**





| nb. | size (width, length, height; volume) | cliff local orientation* | joint sets* | kinematics | ID crackmeter | ID tiltmeter |
|---|---|---|---|---|---|---|
| 3 | 4.0, 2.0, 1.5 m; 12.0 m$^3$ | 235N85° | 100N82°, 05N80°, 90N85° | wedge sliding, toppling | 03F1 | -- |
| 4 | 1.5, 5.0, 1.0 m; 7.5 m$^3$ | 255N88° | 240N35°, 20N85°, 290N15° | planar sliding | 04F1 | 04I1 |
| 5 | 2.0, 2.0, 1.5 m; 6.0 m$^3$ | 275N85° | 100N80°, 20N75°, 230N75° | wedge sliding, toppling | 05F1, 05F2, 05F3 | -- |
| 16 | 2.0, 2.0, 1.0 m; 4.0 m$^3$ | 330N80° | 05N70°, 350N80°, 205N78° | wedge sliding | 16F1, 16F2 | -- |
| 19 | 5.0, 2.0, 1.5 m; 15.0 m$^3$ | 250N87° | 60N88°, 240N85°, 05N48° | toppling | 19F1, 19F2 | 19I1 |

**Table 1: Geostructural characteristics of the monitored tuff blocks (* orientation is expressed with dip direction-dip values).**



| sensor | parameter | sensitivity | operational range | accuracy | update interval |
|---|---|---|---|---|---|
| crackmeter | displacement | 0,01 mm | -50,0 to 50,0 mm | +/- 0,05 % (-30 to 100 °C) | 30 minutes |
| tiltmeter | angle | 0,01 ° | - 10,0 to 10,0 ° | +/- 1 % (-20 to 80 °C) | 30 minutes |
| thermistor | temperature | 0,1 °C | -30,0 to 70 °C | +/- 2,5 °C (-20 to 70 °C) | 30 minutes |

**Table 2: Technical specifications of the sensors in CC-MoSys cabled with the acquisition unit - model eDAS 16ch (see at https://www.boviar.com/public/pdf/636489456594961401eDAS%2016-32%20CH.pdf).**



| sensor | parameter | data resolution and unit | range | accuracy | sampling rate |
|---|---|---|---|---|---|
| anemometer | wind speed | 0,5 m/s (or 0.97 knots) | 0-89 m/s | +/- 1 m/s or 5% | 3 seconds |
| anemometer | wind direction | 1° | 0-360° | +/- 3° | 3 seconds |
| barometer | barometric pressure | 0,1 hPa | 540 - 1100 hPa | +/- 1 hPa | 1 minute |
| hygrometer | relative humidity | 1 % | 1-100% | +/- 2% | 1 minute |
| rain gauge | rainfall amount | 0,25 mm | 0 – 999,8 mm | +/- 4% | 20-24 seconds |
| rain gauge | rainfall rate | 0.1 mm/h | 0-2438 mm/h | +/- 5% | 20-24 seconds |
| thermometer | air temperature | 0,1°C | - 40,0 to 65°C | +/- 0,3°C | 10-12 seconds |

**Table 3: Technical specifications of the sensors of the used meteorological station - model DAVIS Vantage Pro2 wireless (see at https://www.davisinstruments.com/support/vantage-pro2-wireless-stations/) in DeMSys.**

| parameter code | short name | parameter name | unit | sensor type | tuff block | description of measured parameter |
|---|---|---|---|---|---|---|
| 03F1 | DIS | displacement | [mm] | crackmeter | 03 | displacement measured across crack F1 bounding tuff block nb. 03 during last 30 minutes |
| 04F1 | DIS | displacement | [mm] | crackmeter | 04 | displacement measured across crack F1 bounding tuff block nb. 04 during last 30 minutes |
| 05F1 | DIS | displacement | [mm] | crackmeter | 05 | displacement measured across crack F1 bounding tuff block nb. 05 during last 30 minutes |
| 05F2 | DIS | displacement | [mm] | crackmeter | 05 | displacement measured across crack F2 bounding tuff block nb. 05 during last 30 minutes |
| 05F3 | DIS | displacement | [mm] | crackmeter | 05 | displacement measured across crack F3 bounding tuff block nb. 05 during last 30 minutes |
| 16F1 | DIS | displacement | [mm] | crackmeter | 16 | displacement measured across crack F1 bounding tuff block nb. 16 during last 30 minutes |
| 16F2 | DIS | displacement | [mm] | crackmeter | 16 | displacement measured across crack F2 bounding tuff block nb. 16 during last 30 minutes |
| 19F1 | DIS | displacement | [mm] | crackmeter | 19 | displacement measured across crack F1 bounding tuff block nb. 19 during last 30 minutes |
| 19F2 | DIS | displacement | [mm] | crackmeter | 19 | displacement measured across crack F2 bounding tuff block nb. 19 during last 30 minutes |
| 04I2-X | Angle | angle | [°] | tiltmeter | 04 | variation of horizontal angle measured in tuff block nb. 04 during last 30 minutes |
| 04I2-Y | Angle | angle | [°] | tiltmeter | 04 | variation of vertical angle measured in tuff block nb. 04 during last 30 minutes |
| 04I2-T | T tech | temperature | [°C] | thermistor | 04 | external temperature close to sensor I2 (average on 30 min) |
| 19I3-X | Angle | angle | [°] | tiltmeter | 19 | variation of horizontal angle measured in tuff block nb. 19 during last 30 minutes |
| 19I3-Y | Angle | angle | [°] | tiltmeter | 19 | variation of vertical angle measured in tuff block nb. 19 during last 30 minutes |
| 19I3-T | T tech | temperature | [°C] | thermistor | 19 | external temperature close to sensor 19 (average on 30 min) |
| Temp-1 | TTT | temperature | [°C] | thermistor | -- | external air temperature near acquisition unit (average on 30 min) |

**Table 4: List of measured tuff deformation parameters in CC-MoSys.**

| parameter code | short name | parameter name | unit | sensor type | description of measured parameter |
|---|---|---|---|---|---|
| Temp | TTT | air temperature | [°C] | thermometer | air temperature (average on 10 min) |
| Hum | RH | relative humidity | [%] | hygrometer | relative humidity (average on 10 min) |
| Wi-spe | ff | wind speed | [m/s] | anemometer | wind speed (average on last 10 min) |
| Hi-Wi-spe | ff gust | gust speed | [m/s] | anemometer | maximum gust speed during last 10 minutes |
| Wi-dir | Wind dir descr | wind direction | [°] | anemometer | prevalent direction of wind during last 10 minutes |
| Hi-Wi-dir | Wind dir descr | gust direction | [°] | anemometer | direction of maximum gust during last 10 minutes |
| Bar | PPPP | barometric pressure | [hPa] | barometer | site atmospheric pressure (average on 10 min) adjusted to sea level atmospheric pressure |
| Rain | Precip | rainfall amount | [mm] | pluviometer | rainfall amount (cumulated on 10 minutes) |
| Rain-rate | Precip | rainfall rate | [mm/h] | pluviometer | maximum instantaneous rainfall rate during last 10 minutes |

**Table 5: List of measured meteorological parameters in DeMSys.**

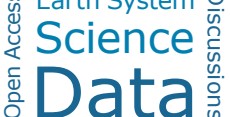

| | Minimum | 1st Quartile | Median | Mean | 3rd Quartile | Maximum | Standard deviation | Num. of measures |
|---|---|---|---|---|---|---|---|---|
| **F03F1** | -0.53 | -0.01 | 0.13 | 0.201 | 0.34 | 1 | 0.302 | 53618 |
| **F04F1** | -2.92 | -1.21 | -0.79 | -0.68 | -0.27 | 2.64 | 0.783 | 29551 |
| **F05F1** | -0.85 | -0.29 | -0.04 | -0.001 | 0.22 | 1 | 0.345 | 55058 |
| **F05F2** | -2.31 | -0.84 | -0.32 | -0.316 | 0.13 | 2 | 0.758 | 55058 |
| **F05F3** | -0.98 | -0.44 | -0.1 | -0.112 | 0.14 | 1 | 0.382 | 55058 |
| **F16F1** | -0.29 | 0.09 | 0.15 | 0.16 | 0.24 | 0.43 | 0.091 | 55058 |
| **F16F2** | -0.28 | -0.08 | 0 | 0.02 | 0.13 | 0.2 | 0.113 | 1339 |
| **F19F1** | -0.79 | -0.29 | -0.17 | -0.142 | -0.02 | 0.39 | 0.182 | 55058 |
| **F19F2** | -0.84 | -0.28 | -0.17 | -0.151 | -0.04 | 0.39 | 0.177 | 55058 |
| **F04I2-X** | -0.62 | -0.28 | -0.19 | -0.179 | -0.04 | 0.08 | 0.137 | 55058 |
| **F04I2-Y** | -0.26 | 0.2 | 0.27 | 0.234 | 0.31 | 0.53 | 0.129 | 55058 |
| **F19I3-X** | -0.53 | -0.25 | -0.2 | -0.205 | -0.15 | 0.06 | 0.082 | 55058 |
| **F19I3-Y** | -0.33 | 0.03 | 0.1 | 0.112 | 0.21 | 0.41 | 0.119 | 55058 |
| **F04I2-T** | -0.01 | 13.65 | 19.04 | 19.18 | 24.23 | 42.79 | 6.844 | 55058 |
| **F19I3-T** | -1.36 | 12.5 | 17.72 | 17.84 | 22.93 | 38.58 | 6.584 | 55058 |
| **Temp-1** | 0.86 | 13.6 | 18.6 | 18.54 | 23.5 | 43.86 | 6.012 | 55058 |
| **Temp \*** | -2.8 | 12.6 | 16.5 | 17.27 | 22.27 | 34.20 | 5.877 | 79473 |

**Table 6: Descriptive statistics of tuff deformation data and temperatures measured by CC-MoSys. \* "Temp" data, measured by DeMSyS from 1 Jan. 2013 to 31 Dec. 2018, have been aggregated to 30 minutes for comparison with CC-MoSys temperature data.**



| | Minimum | 1st Quartile | Median | Mean | 3rd Quartile | Maximum | Standard deviation | num. of measures |
|---|---|---|---|---|---|---|---|---|
| **Temp** | -2.9 | 12.6 | 16.5 | 17.2 | 22.1 | 33.8 | 5.8 | 238422 |
| **Hum** | 23.0 | 64.0 | 75.0 | 72.8 | 83.0 | 98.0 | 13.4 | 238418 |
| **Bar** | 982.7 | 1011.9 | 1015.6 | 1015.8 | 1019.7 | 1036.8 | 6.9 | 238793 |
| **Rain** | 0.0 | 0.0 | 0.0 | 0.0 | 0.0 | 17.6 | 0.2 | 238500 |
| **Rain_rate** | 0.0 | 0.0 | 0.0 | 0.3 | 0.0 | 292.6 | 4.3 | 238500 |
| **Wi_speed** | 0.0 | 1.7 | 3.5 | 4.4 | 6.1 | 40.9 | 3.5 | 200945 |
| **HiWi_spe** | 0.0 | 4.3 | 7.8 | 8.7 | 11.3 | 61.7 | 6.0 | 200945 |

**Table 7: Descriptive statistics of temperature and meteorological data measures by DeMSyS.**





|  | **2014** | **2015** | **2016** | **2017** | **2018** |
|---|---|---|---|---|---|
| Avg. Wind Speed | 2.3 | 2.2 | 2.5 | 2.1 | 2.3 |
| Dominant direction | SSE | SSE | SSE | SSE | SSE |
| Maximun gust | 23.7 | 27.8 | 31.8 | 22.8 | 28.2 |

**Table 8: Annual average values of wind measures during 2014-2018.**





|  | JAN | FEB | MAR | APR | MAY | JUN | JUL | AUG | SEP | OCT | NOV | DEC |
|---|---|---|---|---|---|---|---|---|---|---|---|---|
| Avg.Wind Speed | 2,8 | 2,8 | 2,5 | 2,2 | 2,0 | 1,9 | 1,7 | 1,5 | 2,0 | 2,2 | 2,5 | 2,1 |
| Dominant direction | WSW | SSE | SSE, WSW | WSW, SSE | NE, SE | W, SE, NE | SW, NE | SW, NE | WSW, SSE | WSW, NNE | WSW, NE, SE | SSE |
| Maximun gust | 25.5 | 31.8 | 27.8 | 17.5 | 15.7 | 21.5 | 19.7 | 16.1 | 18.8 | 28.2 | 27.3 | 22.4 |

**Table 9: Monthly average values of wind measures during 2014-2018.**



|  | Min. | 1st. Qu. | Median | Mean | 3rd Qu. | Max | St. dev. | Var. | NA's |
|---|---|---|---|---|---|---|---|---|---|
| **F03F1r** | -0.17 | -0.01 | 0.14 | 0.20 | 0.34 | 1.23 | 0.30 | 0.09 | 657 |
| **F04F1r** | -2.10 | -1.13 | -0.79 | -0.68 | -0.36 | 1.42 | 0.73 | 0.54 | 1150 |
| **F05F1r** | -0.51 | -0.29 | -0.03 | 0.00 | 0.21 | 1.11 | 0.34 | 0.12 | 628 |
| **F05F2r** | -1.83 | -0.84 | -0.32 | -0.31 | 0.13 | 1.83 | 0.76 | 0.58 | 628 |
| **F05F3r** | -0.69 | -0.45 | -0.10 | -0.11 | 0.15 | 1.07 | 0.38 | 0.15 | 628 |
| **F16F1r** | -0.03 | 0.09 | 0.15 | 0.16 | 0.24 | 0.40 | 0.09 | 0.01 | 628 |
| **F16F2r** | -0.13 | -0.06 | 0.06 | 0.03 | 0.13 | 0.17 | 0.11 | 0.01 | 1739 |
| **F19F1r** | -0.46 | -0.29 | -0.17 | -0.14 | -0.02 | 0.37 | 0.18 | 0.03 | 628 |
| **F19F2r** | -0.49 | -0.29 | -0.17 | -0.15 | -0.03 | 0.35 | 0.18 | 0.03 | 628 |
| **F04I2_Xr** | -0.57 | -0.28 | -0.18 | -0.18 | -0.04 | 0.03 | 0.13 | 0.02 | 628 |
| **F04I2_Yr** | -0.11 | 0.21 | 0.27 | 0.24 | 0.31 | 0.43 | 0.13 | 0.02 | 628 |
| **F19I3_Xr** | -0.48 | -0.25 | -0.20 | -0.21 | -0.15 | -0.01 | 0.08 | 0.01 | 628 |
| **F19I3_Yr** | -0.14 | 0.03 | 0.10 | 0.11 | 0.22 | 0.34 | 0.12 | 0.01 | 628 |
| **F04I2_T** | 2.16 | 13.77 | 19.34 | 19.20 | 24.71 | 30.79 | 6.07 | 36.83 | 628 |
| **F19I3_T** | 1.06 | 12.66 | 17.73 | 17.86 | 23.39 | 29.51 | 5.95 | 35.37 | 628 |
| **Temp_1** | 2.41 | 13.63 | 18.35 | 18.56 | 23.60 | 30.36 | 5.65 | 31.95 | 628 |
| **Temp** | -0.31 | 12.53 | 16.70 | 17.26 | 22.20 | 30.10 | 5.71 | 32.60 | 174 |
| **Hum** | 36.50 | 65.02 | 75.20 | 72.79 | 81.37 | 96.11 | 11.26 | 126.73 | 174 |
| **Bar** | 987.60 | 1011.80 | 1015.40 | 1015.60 | 1019.40 | 1035.10 | 6.75 | 45.54 | 171 |
| **Wi_speed** | 0.00 | 2.38 | 3.34 | 4.02 | 5.27 | 23.03 | 2.75 | 7.54 | 171 |
| **Wi-P_norm** | 0.00 | 0.43 | 1.65 | 2.79 | 3.21 | 56.25 | 4.83 | 23.30 | 171 |
| **Rain** | 0.00 | 0.00 | 0.00 | 1.81 | 0.40 | 79.00 | 5.24 | 27.50 | 171 |
| **Rain_rate** | 0.00 | 0.00 | 0.00 | 10.07 | 0.00 | 240.00 | 30.66 | 939.94 | 172 |
| **HiWi_spe** | 5.20 | 12.20 | 16.50 | 18.27 | 22.50 | 67.60 | 8.29 | 68.73 | 171 |
| **HiWi_P_norm** | 0.00 | 42.87 | 78.27 | 112.87 | 118.76 | 1400.61 | 135.93 | 18476.16 | 193 |

**Table 10 Descriptive statistics of daily tuff deformation and meteorological data.**



| | F04I2-T | F19I3-T | Temp-1 | Temp | Hum | Bar | Wi-speed | Wi-P_norm | Rain | Rain-rate | HiWi-spe | HiWi-P_norm |
|---|---|---|---|---|---|---|---|---|---|---|---|---|
| **F03F1** | 0.36 | 0.37 | 0.40 | 0.40 | -0.11 | 0.04 | -0.20 | -0.04 | -0.09 | -0.04 | -0.15 | -0.03 |
| **F04F1** | **-0.57** | **-0.57** | **-0.58** | **-0.58** | -0.08 | -0.05 | 0.20 | 0.00 | 0.13 | 0.03 | 0.23 | 0.04 |
| **F05F1** | -0.39 | -0.38 | -0.37 | -0.34 | -0.23 | 0.37 | -0.04 | -0.14 | -0.06 | -0.06 | -0.08 | -0.15 |
| **F05F2** | 0.38 | 0.38 | 0.40 | 0.41 | -0.08 | 0.11 | -0.21 | -0.07 | -0.05 | 0.00 | -0.19 | -0.07 |
| **F05F3** | -0.46 | -0.45 | -0.44 | -0.42 | -0.17 | 0.33 | 0.00 | -0.11 | -0.01 | -0.03 | -0.03 | -0.12 |
| **F16F1** | -0.43 | -0.42 | -0.39 | -0.39 | -0.19 | 0.17 | 0.04 | -0.07 | -0.01 | -0.04 | 0.09 | -0.04 |
| **F16F2** | 0.31 | 0.36 | 0.31 | 0.40 | -0.23 | 0.36 | -0.20 | **-0.56** | -0.25 | -0.09 | -0.42 | **-0.60** |
| **F19F1** | 0.13 | 0.14 | 0.17 | 0.17 | -0.14 | 0.15 | -0.15 | -0.05 | -0.07 | -0.04 | -0.17 | -0.07 |
| **F19F2** | 0.15 | 0.16 | 0.18 | 0.19 | -0.16 | 0.18 | -0.16 | -0.06 | -0.11 | -0.07 | -0.20 | -0.09 |
| **F04I2-X** | 0.07 | 0.08 | 0.07 | 0.09 | 0.01 | 0.13 | 0.05 | 0.04 | -0.01 | 0.02 | -0.10 | -0.03 |
| **F04I2-Y** | 0.42 | 0.43 | 0.43 | 0.41 | -0.01 | 0.05 | -0.19 | -0.04 | -0.09 | -0.02 | -0.16 | -0.04 |
| **F19I3-X** | **0.68** | **0.69** | **0.69** | **0.73** | -0.04 | 0.08 | -0.22 | -0.05 | -0.13 | -0.05 | -0.30 | -0.10 |
| **F19I3-Y** | **0.62** | **0.62** | **0.64** | **0.65** | 0.02 | -0.01 | -0.27 | -0.05 | -0.11 | -0.03 | -0.22 | -0.05 |

**Table 11. Correlation matrix of tuff deformation and meteorological data; the correlations greater than |0.5| or |0.33| are highlighted.**

|  | **Temp** | **Temp-7** | **Temp-14** | **Temp-21** | **Temp-28** | **Temp-35** | **Temp-42** | **Temp-49** | **Temp-56** | **Temp-63** |
|---|---|---|---|---|---|---|---|---|---|---|
| **F03F1** | 0.40 | 0.43 | 0.43 | 0.44 | 0.44 | 0.42 | 0.40 | 0.39 | 0.36 | 0.33 |
| **F04F1** | 0.75 | 0.71 | 0.70 | 0.69 | 0.67 | 0.64 | 0.60 | 0.55 | 0.51 | 0.47 |
| **F05F1** | -0.34 | -0.25 | -0.18 | -0.11 | -0.04 | 0.01 | 0.07 | 0.15 | 0.21 | 0.25 |
| **F05F2** | 0.41 | 0.47 | 0.50 | 0.54 | 0.57 | 0.58 | 0.59 | 0.60 | 0.60 | 0.59 |
| **F05F3** | -0.42 | -0.33 | -0.25 | -0.18 | -0.11 | -0.05 | 0.01 | 0.09 | 0.16 | 0.22 |
| **F16F1** | -0.39 | -0.30 | -0.27 | -0.25 | -0.22 | -0.20 | -0.18 | -0.14 | -0.11 | -0.08 |
| **F16F2** | 0.40 | 0.72 | 0.63 | 0.83 | 0.80 | 0.91 | 0.35 | 0.50 | 0.96 | 0.70 |
| **F19F1** | 0.17 | 0.22 | 0.24 | 0.27 | 0.29 | 0.28 | 0.28 | 0.30 | 0.30 | 0.30 |
| **F19F2** | 0.19 | 0.25 | 0.26 | 0.28 | 0.30 | 0.29 | 0.29 | 0.30 | 0.31 | 0.30 |
| **F04I2-X** | 0.09 | 0.09 | 0.09 | 0.11 | 0.12 | 0.12 | 0.13 | 0.14 | 0.17 | 0.19 |
| **F04I2-Y** | 0.41 | 0.40 | 0.42 | 0.43 | 0.43 | 0.44 | 0.43 | 0.41 | 0.39 | 0.35 |
| **F19I3-X** | 0.73 | 0.68 | 0.65 | 0.66 | 0.65 | 0.63 | 0.59 | 0.56 | 0.54 | 0.49 |
| **F19I3-Y** | 0.66 | 0.61 | 0.61 | 0.61 | 0.59 | 0.58 | 0.56 | 0.51 | 0.48 | 0.42 |

**Table 12 Correlation matrix between tuff deformation data and air temperature for different time lags (expressed in numeber of days).**



| | DF | SumSQ | MeanSQ | F |
|---|---|---|---|---|
| **Temp** | 1 | 3.9286 | 3.9286 | 1463.13*** |
| **Wi-P_norm** | 1 | 0.0174 | 0.174 | 6.4817* |
| **Residuals** | 861 | 2.3118 | 0.0027 | |
| **Multiple R-squared** | 0.6306 | **Adjusted R-squared** | 0.6297 | 734.8*** |

**Table 13 Analysis of variance (ANOVA). p values: 0 '***' 0.001 '**' 0.01 '*' 0.05 '.' 0.1 ' ' 1**



|  | Estimate | Standard error |
|---|---|---|
| **(Constant)** | -0.3881*** | 0.0053 |
| **Temp** | 0.0114*** | 0.0003 |
| **Wi-P_norm** | 0.0008* | 0.0003 |

**Table 14 Parameters estimation in the regression model. p values:  0 '***' 0.001 '**' 0.01 '*' 0.05 '.' 0.1 ' ' 1**