# Peer review of "Integrated dataset of deformation measurements in fractured volcanic tuff and meteorological data (Coroglio coastal cliff, Naples, Italy)."

_Earth System Science Data, 2019_

## Referee Comment (RC1) · Anonymous Referee #1 · 11 Oct 2019

Manuscript: Integrated dataset of deformation measurements in fractured volcanic tuff and meteorological data (Coroglio coastal cliff, Naples, Italy).

Overview comments:

This research paper presents a dataset of deformation measurements carried out in five fractured volcanic tuff blocks at the Coroglio coastal cliff (Naples, Italy). Multiple tuff deformation parameters were obtained from crackmeters and tiltmeters during 4 years. These values were correlated with numerous meteorological variables from a weather station in order to detect possible relationships between the two data sets. It is concluded that (1) the deformation mainly increases if temperature increases and (2)

fractures deform both synchronously and in a delayed way with temperature.

The work is well written and illustrated with figures and tables, and their conclusions seem to be consistent with the results of deformation measurements and their correlation with numerous meteorological variables. However, a restructuration of some parts of the paper is considered appropriate (see 'line-by-line commentary'). In some cases, authors included data, methods and results in the same section. Therefore, define independent sections for 'Data and methods' and 'Results' may be appropriate. On the other hand, the number of tables and figures seems excessive. Select the most relevant figures and tables to be included in the manuscript and put the rest in a 'Supplementary information' section.

General vote: manuscript may become acceptable after major changes, which are detailed in this review.

Line-by-line commentary:

Page 1, line 17: Freezing conditions in the southwestern coast of Italy? Page 2, line 6: It may be more appropriate 'mass wasting' instead of 'mass movement'. Page 2, lines 11-18: References indicated in these lines seem to be examples (indicate 'e.g.' before references). Page 2, line 25: 'relicts' instead of 'relics'. Page 3, line 23: It may be better to say 'structural instability'. Page 4, line 7: 0.07 m/yr in that time period? Page 4, line 12: Consider to define a 'Data and methods' section here with sub-sections (e.g., 3.1 The monitoring system, 3.2 Tuff deformation data, and 3.3 Meteorological data). Page 6, line 7: 'max-min value ranges' instead of 'max-min values'. Page 6, lines 7-32 and Page 7, lines 1-8: These lines correspond to 'Results' section. Page 6, line 11: code; 04F1? Page 6, line 18: Add a space between 'F2' and 'sensor'. Page 6, line 26: -0.20° and -0.30°. Page 7, lines 1-29: These lines correspond to 'Data and methods' section. Page 7, line 25: It suggests to define the expression 1 as $Pn = 0.613vn2$. Page 7, line 27: Replace '*' by 'Åů' in the equation (i.e., NÅům2). Page 8, line 8: . . .the difference between the minimum and maximum temperatures ($\Delta T$). Page 8, line 24:

[Figure]

...daily values, probably associated with the proximity to the sea. Page 9, lines 9 and 10: Are these high rain rates associated with late summer storms? Page 10, line 12: The wind rose diagram... Page 10, line 13: ...from the II and IV quadrants of the rose diagram. Page 12, line 11: ...(e.g., 0-0.54...)? Page 12, lines 13-15: Why authors do not use F19I3 too? Page 13, line 7: ...acting on the cliff (e.g., F16F2).

Figures:

Figure 1: Add coordinate grid in Fig. 1a and indicate the source of the orthophoto in Fig.2b.

Figure 2: It would be recommendable to indicate upper, middle and lower parts in this figure.

Figure 3: Add x- and y-axis labels (i.e., Horizontal distance (m) and Elevation (m a.s.l.).

Figures 6-12, 22, 25-30: Add x- and y-axis labels. Define 'a' and 'b' in Fig. 15.

Page 34: Indicate the figure number.

Please also note the supplement to this comment:
https://www.earth-syst-sci-data-discuss.net/essd-2019-139/essd-2019-139-RC1-supplement.pdf

---

## Referee Comment (RC2) · Anonymous Referee #2 · 18 Oct 2019

The manuscript represents an interesting interdisciplinary local case study of a tuff sea cliff. Data are well presented and each to other integrated, as well as discussed with the aim of understanding the possible triggering of rockfalls. Figures, diagrams and tables are congruent and support the conclusions well, although some figures may seem redundant. This manuscript is a good example of monitoring aimed at determining the rockfall hazard of an eroding cliff located in an urbanized area of the periphery of a city.

---

## Referee Comment (RC3) · Anonymous Referee #3 · 18 Oct 2019

This is an interesting paper about the assessment of cliff erosion processes after applying a number of techniques for measuring mainly atmospheric/climatic and deformational variables. The article is well structured and written, and incorporates a big number of data, from which results and conclusions seem very consistent. Perhaps there is a high number of figures; some of them could be considered as Suplementary material and remain a selection of them. In fact, some of the figures seem somewhat repetitive. Anyway, I recommend publication with only minor changes related to small mistakes detected along the text: Page 6, line 4: Table 6; line 12: the beginning of the sentence should better be "Parameters of block 05. . ."; line 15: a reference to Fig. 8 must be included after "16F1 sensor"; line 23 at the end: consider "during the time

to some preferred. . ."; line 25: "shown" instead of "shows"; line 26: "by angle values between. . ." Page 7, line 13: instead of "are more large", consider "are larger". Page 11, line 4: "This is a direct consequence. . ." There is no reference to Fig. 23 along the text. Be careful with the style. Mind the verb tenses. For example, in the second paragraph of page 11 present tense and past perfect alternate without any justification. Conclusions: too brief in my opinion. The last sentence cites the influence of human action and volcano-tectonic activity, which are not considered along in this work. The citation of Furlani et al., 2014 (page 2, line10) is not included in the Reference list. Two references in the list are not cited along the text: Bakun-Mazor et al. (2013), and Fortelli et al. (2018)
* * *

---

## Referee Comment (RC4) · Anonymous Referee #4 · 18 Oct 2019

General comments: It is an interesting paper on the effects of external forcing factors (mainly temperature, rainfall and wind) on the deformation of a rocky cliff, based on in situ measurement of the meteorological and deformation data for a considerable time period (4 years). This provides an important contribution for the knowledge on the generation of cliff instability phenomena. The paper has a clear structure, is well written and deals with large datasets, which enabled consistent, if somewhat scarce, conclusions. Figures 6 to 11 (bar plots) could be grouped, since they are somewhat repetitive, and could be presented with smaller size.

I recommend publication with minor changes.

[Figure]

Page 6, line 8 – Instead of "max-min values" it is better "value ranges"; please indicate units in the numbers. Fig. 6, 7a, 8, 9 10, 11 – Instead "sensors" it would be clearer to indicate "crackmeters".

---

## Author Comment (AC1) · 5 Dec 2019

We would like to thank all reviewers for their positive comments towards our work and their constructive suggestions that aided in improving the manuscript.

Reviewer 1

We thank Reviewer 1 for the encouraging comments and suggested revisions.

> "General comments: This research paper presents a dataset of deformation measurements carried out in five fractured volcanic tuff blocks at the Coroglio coastal cliff (Naples, Italy). Multiple tuff deformation parameters were obtained from crackmeters

and tiltmeters during ∼4 years. These values were correlated with numerous meteorological variables from a weather station in order to detect possible relationships between the two data sets. It is concluded that (1) the deformation mainly increases if temperature increases and (2) fractures deform both synchronously and in a delayed way with temperature. The work is well written and illustrated with figures and tables, and their conclusions seem to be consistent with the results of deformation measurements and their correlation with numerous meteorological variables. However, a restructuration of some parts of the paper is considered appropriate (see 'line-by-line commentary'). In some cases, authors included data, methods and results in the same section. Therefore, define independent sections for 'Data and methods' and 'Results' may be appropriate. On the other hand, the number of tables and figures seems excessive. Select the most relevant figures and tables to be included in the manuscript and put the rest in a 'Supplementary information' section."

Following the reviewer's suggestion, we have rearranged some parts of the paper. Independent sections have been created for 'Data and methods' and 'Results' in the revised manuscript, avoiding confusion between collected data and obtained results. All figures have been upgraded following the reviewer's comments. Moreover, we have reduced the number of figures (from 31 to 25): e.g. see new histograms grouping previous figures from 6 to 9, and 10 - 11, new diagrams grouping previous multi-part figures 14 and 15, and new figures grouping previous figures from 25 to 30 (see attached supplement).

> "Line-by-line commentary: Page 1, line 17: Freezing conditions in the southwestern coast of Italy? Page 2, line 6: It may be more appropriate 'mass wasting' instead of 'mass movement'. Page 2, lines 11-18: References indicated in these lines seem to be examples (indicate 'e.g.' before references). Page 2, line 25: 'relicts' instead of 'relics'. Page 3, line 23: It may be better to say 'structural instability'. Page 4, line 7: 0.07 m/yr in that time period? Page 4, line 12: Consider to define a 'Data and methods' section here with sub-sections (e.g., 3.1 The monitoring system, 3.2 Tuff deformation

data, and 3.3 Meteorological data). Page 6, line 7: 'max-min value ranges' instead of 'max-min values'. Page 6, lines 7-32 and Page 7, lines 1-8: These lines correspond to 'Results' section. Page 6, line 11: code; 04F1? Page 6, line 18: Add a space between 'F2' and 'sensor'. Page 6, line 26: -0.20° and -0.30°. Page 7, lines 1-29: These lines correspond to 'Data and methods' section. Page 7, line 25: It suggests to define the expression 1 as Pn = 0.613vn2. Page 7, line 27: Replace '*' by 'Âů' in the equation (i.e., NÂům2). Page 8, line 8: . . .the difference between the minimum and maximum temperatures ($\Delta$T). Page 8, line 24: ...daily values, probably associated with the proximity to the sea. Page 9, lines 9 and 10: Are these high rain rates associated with late summer storms? Page 10, line 12: The wind rose diagram. . . Page 10, line 13: . . .from the II and IV quadrants of the rose diagram. Page 12, line 11: ... (e.g., 0-0.54. . .)? Page 12, lines 13-15: Why authors do not use F19I3 too? Page 13, line 7: . . .acting on the cliff (e.g., F16F2)."

We have made all corrections requested in the reviewer's line-by-line comments. More in detail, referring to Page 1, line 17 comment, temperature data show that freezing conditions very rarely occur in the study area, so the text has been changed in "Among the triggering mechanisms, the most relevant are related to meteorological factors, such as precipitation and thermal expansion due to solar heating of rock surfaces." Referring to Page 3, line 23 comment, as the instability on the cliff is due to both structural and geomorphological causes we suggest to use a more general definition as "cliff instability". Referring to Page 9, lines 9-10 comment, these high rain rates are usually associated with late summer storms. Referring to Page 12, lines 13-15 comment, in this section (lines 13-15) we are analysing the correlations between the variables measuring temperature and the variables measuring the wind pressure, so the tiltmeter data (F19I3) are not considered here.

> "Figures: Figure 1: Add coordinate grid in Fig. 1a and indicate the source of the or-thophoto in Fig.2b. Figure 2: It would be recommendable to indicate upper, middle and lower parts in this figure. Figure 3: Add x- and y-axis labels (i.e., Horizontal distance

(m) and Elevation (m a.s.l.). Figures 6-12, 22, 25-30: Add x- and y-axis labels. Define 'a' and 'b' in Fig. 15. Page 34: Indicate the figure number."

We have made all corrections requested in the Figures comments. All figures have been redrawn. Older figures 6 to 9, 10-11, 18, 25-26, 27-28 and 29-30 have been grouped or deleted.

Reviewer 2

We thank Reviewer 2 for the valuable comments and suggestions.

> "The manuscript represents an interesting interdisciplinary local case study of a tuff sea cliff. Data are well presented and each to other integrated, as well as discussed with the aim of understanding the possible triggering of rockfalls. Figures, diagrams and tables are congruent and support the conclusions well, although some figures may seem redundant. This manuscript is a good example of monitoring aimed at determining the rockfall hazard of an eroding cliff located in an urbanized area of the periphery of a city."

Following the reviewer's suggestion, we have reduced the number of figures (from 31 to 25). Older figures 6 to 9, 10-11, 18, 25-26, 27-28 and 29-30 have been grouped or deleted (see attached supplement).

Reviewer 3

We thank Reviewer 3 for the fulfilling comments and the suggested revisions.

> "This is an interesting paper about the assessment of cliff erosion processes after applying a number of techniques for measuring mainly atmospheric/climatic and deformational variables. The article is well structured and written, and incorporates a big number of data, from which results and conclusions seem very consistent. Perhaps there is a high number of figures; some of them could be considered as Supplementary material and remain a selection of them. In fact, some of the figures seem somewhat repetitive. Anyway, I recommend publication with only minor changes related to small

mistakes detected along the text: Page 6, line 4: Table 6; Page 6, line 12: the beginning of the sentence should better be "Parameters of block 05 ..."; Page 6, line 15: a reference to Fig. 8 must be included after "16F1 sensor"; Page 6, line 23 at the end: consider "during the time to some preferred ..."; Page 6, line 25: "shown" instead of "shows"; Page 6, line 26: "by angle values between ..."; Page 7, line 13: instead of "are more large", consider "are larger"; Page 11, line 4: "This is a direct consequence ..."; There is no reference to Fig. 23 along the text. Be careful with the style. Mind the verb tenses. For example, in the second paragraph of page 11 present tense and past perfect alternate without any justification; Conclusions: too brief in my opinion. The last sentence cites the influence of human action and volcano-tectonic activity, which are not considered along in this work. The citation of Furlani et al., 2014 (page 2, line10) is not included in the Reference list. Two references in the list are not cited along the text: Bakun-Mazor et al. (2013), and Fortelli et al. (2018)."

Following the reviewer's suggestion, we have reduced the number of figures (from 31 to 25). Older figures 6 to 9, 10-11, 18, 25-26, 27-28 and 29-30 have been grouped or deleted (see attached supplement). All requested corrections have been added to the text. We have checked and corrected the verb tenses in section 5. Conclusions have been extended and modified. In the last sentence, the reference to the influence of human action and volcano-tectonic activity has been deleted because these factors are not discussed in our manuscript.

Reviewer 4

We thank Reviewer 4 for the suggested revisions and the thorough comments.

> "It is an interesting paper on the effects of external forcing factors (mainly temperature, rainfall and wind) on the deformation of a rocky cliff, based on in situ measurement of the meteorological and deformation data for a considerable time period (4 years). This provides an important contribution for the knowledge on the generation of cliff instability phenomena. The paper has a clear structure, is well written and deals with

large datasets, which enabled consistent, if somewhat scarce, conclusions. Figures 6 to 11 (bar plots) could be grouped, since they are somewhat repetitive, and could be presented with smaller size. Page 6, line 8 – Instead of "max-min values" it is better "value ranges"; please indicate units in the numbers. Fig. 6, 7a, 8, 9 10, 11 – Instead "sensors" it would be clearer to indicate "crackmeters"."

Following the reviewer's suggestion, we have reduced the number of figures (from 31 to 25). Older figures 6 to 9, 10-11, 18, 25-26, 27-28 and 29-30 have been grouped or deleted (see attached supplement). All requested corrections have been made in the text. Conclusions have been modified and extended.

Please also note the supplement to this comment:
https://www.earth-syst-sci-data-discuss.net/essd-2019-139/essd-2019-139-AC1-supplement.pdf

---

## Referee Report (RR1)

**Revised manuscript:** *Integrated dataset of deformation measurements in fractured volcanic tuff and meteorological data (Coroglio coastal cliff, Naples, Italy).*

**Overview comments:**

Authors have implemented all suggestions and changes indicated by referees in their revised manuscript and have answered the formulated questions.

General vote: the revised manuscript should be accepted as is.